

# Quasi-Biennial Oscillation Modulation of Global Monsoon Systems and Regional Teleconnections

Vinay Kumar[1#], Shigeo Yoden[2], Matthew H. Hitchman[3], Tetsuya Takemi[4], and Kosuke Ito[4]

[1]Radio and Atmospheric Physics Lab, Rajdhani College, University of Delhi, New Delhi, 110 015, India
[2]Professor Emeritus of Kyoto University, Kyoto, 606-8501, Japan
[3]Department of Atmospheric and Oceanic Sciences, University of Wisconsin–Madison, Madison, WI, 53706, USA
[4]Disaster Prevention Research Institute, Kyoto University, Kyoto, 611-0011, Japan

[#]*Correspondence to*: Vinay Kumar (dabas.vinay@gmail.com)

**Abstract.** This study investigates the influence of the stratospheric Quasi-biennial oscillation (QBO) on global monsoon systems and their extratropical teleconnections using 42 years (1979–2020) of monthly mean ERA-5 reanalyses data. QBO modulation of precipitation and circulation variables is examined in the context of downward-progressing QBO phase, separately for JJA and DJF. It is found that QBO teleconnections act primarily by modulating regional circulations. During JJA, QBO westerly (W) at 50hPa coincide with an intensified Pacific Walker Circulation, with enhanced rainfall over the Maritime Continent and less over the Western Pacific. In the Northen Hemisphere subtropics for same QBO W, the anticyclonic lower tropospheric circulation in the northwest Pacific is reduced, with rainfall south of Japan shifting eastward. During DJF, QBO teleconnections with the North Atlantic Oscillation (NAO) and the Pacific North America (PNA) pattern are observed. QBO W at 50hPa promotes a positive NAO, which enhances the anticyclonic circulation associated with the Azores High, resulting in a stronger North Atlantic jet stream and a westward shift in precipitation toward the east coast of North America. As QBO phase progress, QBO W at 70 hPa promotes a positive PNA phase, with a trough over the Gulf of Alaska and ridge over the eastern Pacific. This promotes northeastward flow into Alaska and above average precipitation, with less along the west coast of the United States. In each of these examples, the opposite effect is observed during QBO easterly (E) relative to QBO W, suggesting the dominance of linearity in QBO modulations.

## 1. Introduction

The global monsoon (GM) is a large-scale circulation system which helps to control the distribution of precipitation in low- and mid-latitudes. It responds to the annual cycle of solar forcing and is influenced by continental-scale thermal contrast between land and ocean (e.g., Chang et al. Eds., 2005; Krishnamurti et al., 2013). In the GM system, specific features of the underlying surface, including the land-ocean distribution, topography, and oceanic circulation play key roles in contributing to regional differences observed among monsoon systems (Wang and Ding, 2008; Wang et al., 2017). Based on the timing of the wet season, the summer GM can be classified into two main modes: the boreal summer monsoon (June, July, and August, hereinafter JJA) in the Northen Hemisphere (NH) subtropics, and the austral summer monsoon (December, January, and February, hereinafter DJF) in the Southern Hemisphere (SH) subtropics (An et al., 2015). Further, at the regional scale, the





summer GM is categorized in six dominant monsoon systems in the world: North African, Asian, and North American in NH during JJA, and South African, Australian, and South American in SH during DJF. These regional monsoons are not

independent but have a dynamical linkage among them via prevailing wind patterns (Meehl, 1987).

In the past few decades there has been a series of research publications and review articles on the GM (e.g., Chang et al. Eds. 2005; 2011; 2016; 2020; Yoden et al., 2023). However, the GM is a complex system, and its variability across different time scales has a profound societal and economic impact on nearly two-thirds of the world's population. Thus, studying variations in the GM system across subseasonal, seasonal, interannual, and decadal timescales is essential to improve

knowledge and forecasting skill. Forecasting variations in GM dynamics across different time scales remains challenging, due to the highly complex characteristics of teleconnections with various atmospheric circulations. Understanding interannual variability of the GM system presents a significant challenge to the scientific community. An improvement in understanding would directly benefit society through an improved ability to forecast regional and seasonal climate anomalies.

On interannual time scales, the GM system is also influenced by the El Niño Southern Oscillation (ENSO), a major

driver of global teleconnections (Mooley and Parthasarathy, 1984; Shen and Lau, 1995; Krishnamurthy and Goswamy, 2000; Yu et al., 2021). The impact of ENSO can be confined to a specific monsoon system and may vary with the time scale. Recently, Yu et al. (2021) demonstrated that, at interannual time scales, the Indian summer monsoon exhibits a stronger relationship with ENSO, whereas on the decadal time scale, this relationship is weaker. Despite extensive studies on the teleconnection between GM dynamics and surface atmospheric circulations, interannual variability of GM is still unclear, and

further research is needed to improve understanding.

Vertical coupling with stratospheric phenomena can also influence the GM system. The quasi-biennial oscillation (QBO) is a regular dynamical feature of the equatorial stratosphere in the layer ~ 10-100 hPa, in which alternating layers of easterly (E) and westerly (W) winds descend in time, with periodicity ~ 24-32 months (Baldwin et al., 2001). The influence of the QBO is not confined to the tropical stratosphere, but can extend down to the surface across the entire globe through three

primary pathways, by directly affecting tropical convection, by affecting the interaction between subtropical jets (STJs) and synoptic waves, and via the winter stratosphere (Haynes et al., 2021; Kumar et. al., 2022, 2024). The tropical and subtropical pathways are caused by QBO modulation of the zonal mean temperature structure in the upper troposphere and lower stratosphere (UTLS) by QBO mean meridional circulation (MMC) cells, where descending QBO W is associated with a warm anomaly in the tropics and simultaneous cold anomalies in the subtropical UTLS of both hemispheres (Hitchman et al. 2021).

The stratospheric pathway involves QBO modulation of planetary Rossby wave propagation by changing the stratospheric QBO MMC and position of the zero-wind line. This pathway is known as "Holton-Tan effect", or H-T effect (Holton and Tan 1980; 1982), which operates during boreal winter when westerlies exist in the polar stratosphere, QBO E favors a disrupted polar vortex, and therefore a weak Northern Annular Mode (NAM) and increased midlatitude surface cold air outbreaks (e.g., Kumar et al., 2022). In the subtropical route, when a QBO MMC arrives at the tropopause, it causes

antiphased temperature anomalies in the tropics and subtropics, which alters the meridional temperature gradient, hence the distribution of zonal wind, by the thermal wind law. The resulting QBO zonal wind anomalies can influence the STJs, which





can, in turn, interact with synoptic and and planetary-scale eddies which originate in the extratropics and dissipate in the subtropics (Garfinkel and Hartmann, 2011; Inoue and Takahashi, 2013; Haynes et al., 2021). In the tropical route, dynamical and thermodynamic modulation of the tropical UTLS by the QBO MMC is directly linked to the surface region through

modulation of deep convection and its organization (Gray et al., 1992a, 1992b; Collimore et al., 1998, 2003; Kumar et al., 2014; Hitchman et al., 2021; Haynes et al., 2021). QBO modulation of deep convective centers can modulate the planetary wave trains which emanate from them, with poleward energy dispersion along the UTLS. This can, in turn, modulate regional circulation features in the extratropics, including the North Atlantic Oscillation (NAO) and Pacific North America (PNA) pattern.

QBO modulation of deep convection in the tropical route may play a substantial role in tropical rainfall and global circulation, potentially influencing forecasts of weather around the globe (Collimore et al., 2003). Gray et al. (2018) observed increased precipitation in the tropical western Pacific when there is QBO W at 70 hPa, particularly during boreal summer. They also noted that the band of precipitation across the Pacific, associated with the intertropical convergence zone (ITCZ), shifts southward. In the subtropical route, the QBO may directly influence rainfall in the subtropics. For example, when QBO

E occurs at 70 hPa, the East Asian STJs weakens and shifts poleward, weakening the East Asian winter monsoon (Ma et al., 2021).

The effects of the QBO on rainfall in different parts of the globe, particularly within the tropical and subtropical regions, have been examined in some previous observational studies (Seo et al., 2013; Gray et al., 2018; Ma et al., 2021). In addition, several numerical model studies have been conducted to explore modulation rainfall patterns by the QBO (Goswami,

1998; Giorgetta et al., 1999; Brönnimann et al., 2016). However, some contradictions have been observed among the numerical model studies. In a general circulation model (GCM) experiment, Giorgetta et al. (1999) found that the boreal summer monsoon JJA is significantly influenced by the QBO, with the less precipitation over the western Pacific during QBO W but more over the Indian Ocean. On the other hand, Brönnimann et al. (2016) did not find significant QBO effects on the Indian monsoon in either observations or coupled ocean–atmosphere–chemistry climate model simulations.

Recently Yoden et al. (2023) reported on new observational aspects of QBO modulation of the GM system, highlighting modulation of low-pressure cyclonic perturbations over the NH western Pacific during JJA and eastern Pacific during DJF. However, this study does not provide an in-depth view of the QBO association with the GM system from both a phenomenological and mechanical perspective. In a very recent study, Kumar et al. (2024) demonstrated teleconnections between the QBO and regional surface climate in Eurasia and North America during boreal winter. They proposed a new

teleconnection pathway via the UTLS region to the high-latitude surface, independent of the H-T mechanism. Both studies suggest that the role of the QBO in modulating surface weather regimes from the equatorial regions to the polar regions cannot be ignored. Therefore, further systematic work is needed to diagnose the QBO teleconnection pathways, which vary on a seasonal and regional basis and by phase of the QBO.

Also, the ENSO events directly influence the QBO MMC, tropical, subtropical and polar routes (Kumar et al., 2022).

The new teleconnection pathway observed by Kumar et al. (2024) was found to exist only during the neutral phase of ENSO





(neither El Niño or La Niña). Yoden et al. (2023) also excluded extreme El Niño and La Niña events to avoid contamination from them. Therefore, in this study, we will focus on the neutral ESNO period to extract the QBO effects on GM system and regional circulation patterns, focusing on regions from tropical to extratropical latitudes (60°S-60°N) where monsoon circulations have a strong influence on precipitation and wind patterns. Potential routes by which the QBO connects with the GM system will be investigated, focussing on changes associated with specific QBO phases. Separate analyses are presented for both the JJA and DJF seasons. This study examines not only precipitation and its proxy data but also circulation fields. The sensitivity of QBO surface teleconnection modulation to the downward-propagating QBO phase is investigated by systematically varying the altitude of QBO index to search for prominent remote influences of the QBO on the GM system.

## 2. Data and methodology

### 2.1 Data

Monthly mean ERA-5 reanalysis data for the 42-year period 1979 - 2020 (satellite era) are used to analyze the various meteorological variables such as zonal and meridional wind (U, V), vertical wind (W), divergence of horizontal wind, specific humidity (q), and mean sea level pressure (MSLP). ERA-5 offers several improvements over its preceding ERA-Interim version (Hersbach et al., 2020) as it is benefitted from a decade of developments in core dynamics, and model physics. The assimilation of a much larger number of reprocessed datasets has improved ERA-5 reanalysis products which is considerable in the troposphere. In this study, a double prime superscript for a given meteorological variable, $X''$, represents its deviation from the 42-years climatological mean for that month. Outgoing longwave radiation (OLR) monthly data are taken from the National Oceanic and Atmospheric Administration (NOAA) (Gruber and Krueger 1984). To study the monsoon patterns, precipitation (P) data are used from the Global Precipitation Climatology Project (GPCP) version 2.3. This merged data set incorporates precipitation estimates from surface rain gauge stations, low-orbit satellite microwave data, and geostationary satellite infrared data, on a $2.5° \times 2.5°$ (latitude × longitude) global grid (Adler et al., 2003; 2018). This dataset provides a definitive description of global monsoon rainfall patterns over both land and ocean.

SST data were obtained from the Hadley Centre Global Sea Ice and Sea Surface Temperature (HadISST) v1.1 (Rayner et al., 2003) and are used to calculate the monthly Niño 3.4 index for identifying extreme ENSO events. These events are defined using de-seasonalized data over a 42-year period in the Niño 3.4 region (5°N–5°S, 120°W–170°W). Any month is considered to be an extreme El Niño or La Niña period whenever the Niño 3.4 index exceeds the threshold values ±1.0 K (+: El Niño; –: La Niña); we obtained 395 neutral, 57 El Niño, and 52 La Niña months. This study focuses on the 395 neutral-to-moderate months only, and all the analysis is conducted at $2.5° \times 2.5°$ resolution, spanning 60°S to 60°N.

### 2.2 Methodology

A QBO state is defined in a two-dimensional phase space using the first two principal components (*PC1* and *PC2*) of the de-seasonalized zonal mean zonal wind variations in the equatorial lower stratosphere between 10 hPa to 70 hPa (Wallace et al., 1993). Figure S1 in supportive information (hereinafter SI) shows the vertical structure of the first two leading EOFs,




along with scatter plots in the *PC1* and *PC2* phase space for all four seasons: DJF, March, April, and May (MAM), JJA, and

September, October, and November (SON). The QBO state is represented by the phase angle $\theta = tan^{-1}\left(\frac{PC2'}{PC1'}\right)$, where

*PC1'* and *PC2'* are the principal components with respect to the centroid of all data points (black + marker in Fig. S1). See

Kumar et al. (2022, 2024) and Yoden et al. (2023) for a detailed description of the methodology used in employing EOF

analysis to define QBO phase. In this study we divide QBO phase into the following eight 45° angular bins: phase 1 (0 – 45°),

phase 2 (45 – 90°), phase 3 (90 – 135°), phase 4 (135 – 180°), phase 5 (180 – 225°), phase 6 (225 – 270°), phase 7 (270 –

315°), and phase 8 (315 – 360°), hereinafter designated P1 to P8 (Hitchman et al., 2021; Yoden et al., 2023).

140         Figure 1 shows vertical profiles of the composite means of de-seasonalized zonal mean zonal wind at the equator, for

each of the eight phases separately, during JJA (a-d) and DJF (e-h). Each subplot contains two profiles corresponding to

opposite QBO phases at the central angle $\theta_c$ (i.e., $\theta_c$, and $\theta_c + 180°$), shown in red for P1 to P4 and blue for P5 to P8. The

composite difference between the opposite phases is shown as a black dashed line (Fig. 1). An open circle on the right y-axis

indicates that the composite difference for a given level is statistically significant, using a two-sided Student's *t*-test with two

independent samples in each phase. Successive downward phase propagation of QBO W (QBO E) phase can be seen from P1

to P4 (P5 to P8) between 20 hPa and 70 hPa. Most of the major features of QBO propagation are similar in the two seasons

for each phase pair (Fig. 1).

        When QBO winds maximize at 70 hPa, the profile of P1-P5 is different above 0.3 hPa for JJA (Fig. 1a) and DJF (Fig.

1e) and the magnitude of P1-P5 in the lowest stratosphere is diminished during DJF. At 70 hPa, the whiskers are clearly

separated in the JJA profile (Fig. 1a), but overlap in the DJF profile (Fig. 1e). Kumar et al. (2013) showed that the tropopause

height is at a higher altitude (~ 90 hPa) during DJF compared to JJA (~ 110 hPa), suggesting that the higher tropopause during

DJF may be inhibiting deeper penetration of the QBO signal to the troposphere. Note that DJF is a dynamically active season

in the NH (and during SON in the SH), with westerly winds permitting upward propagation of planetary waves into the

stratosphere and the possibility of the H-T mechanism. Also, the deep convection penetrates higher over Indonesia during DJF,

the cloud tops would "poke up higher" into QBO thermal anomalies associated with the MMC, which can modulate upper

tropospheric instability in deep convection. From this mechanism, one might expect the QBO effects during DJF for the

tropical route. On another hand, during JJA, a lack of stratospheric westerlies implies that the teleconnection likely occurs

along the subtropical UTLS. So, seasonality is an important factor in understanding QBO dynamical teleconnections.





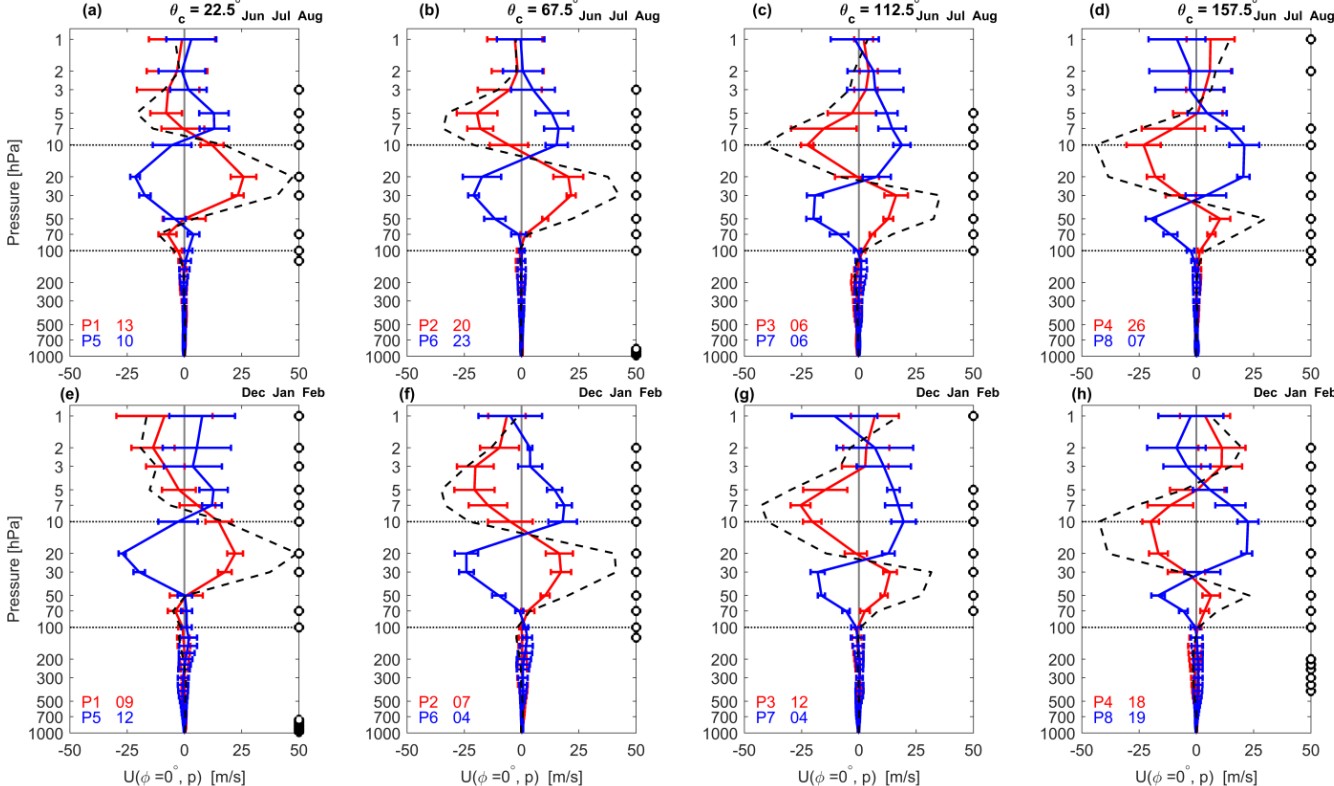

**Figure 1.** Average vertical profiles of zonal mean zonal wind **U″** anomalies for each of the eight phases P1 to P4 (red) and P5 to P8 (blue), along with composite differences between these QBO opposite phases (black dashed line) for JJA (a-d) and DJF (e-h). The whiskers denote ± one standard deviation for each composite phase, and an open circle is plotted on the right y-axis if the composite difference at that level exceeds 90% statistical significance. The total number of samples for each phase is written at the bottom left corner in each panel.

Based on our previous survey of modulations of the GM system by QBO phase (Yoden et al., 2023), this study will focus on the phases when QBO anomalies arrive in the UTLS regions, separately for JJA and DJF, with results presented for the composite difference P4 – P8 (QBO W – QBO E 50 hPa), as well as for P5 – P1 (QBO W – QBO E at 70 hPa). For an easy understanding of the QBO signature in the GM system, it is essential to grasp the fundamental dynamics of the GM system; therefore, we will first discuss its climatology in the next section.

## 3. Climatological cycle of the GM

Figure 2 provides an overview of the 42-year zonal mean and non-zonal climatology of the GM system in precipitation (P), together with the wind patterns (U, V) at 950 hPa, and presents the annual, boreal summer monsoon JJA, and austral summer monsoon DJF mean structures separately. This climatology is created using only neutral ENSO months. The 1st column represents the zonal mean structure of [U] (solid line) and [V] (dashed line). The 2nd column shows zonal mean precipitation [P], while the 3rd column shows a longitude-latitude section of precipitation (P) along with arrows representing



(U, V) wind patterns. From top to bottom, rows show the mean states for all year, JJA, DJF, and the composite difference JJA – DJF. Note that a positive value (golden color) in the composite difference panel denotes the region of heavy precipitation during JJA in comparison to DJF, and a negative value (green color) denotes the region of heavy precipitation during DJF compared with JJA. This figure is an updated version of Fig. 3 of Yoden et al. (2023), showing a wider latitudinal domain, from 60°S to 60°N, and including wind patterns in the boundary layer at 950 hPa.

**Figure 2.** ERA-5 climatology (1979-2020) of precipitation (P) and horizontal winds (U, V) at 950 hPa, using only neutral ENSO months. The 1st and 2nd columns show latitudinal profiles of zonal mean zonal [U] (solid line) and meridional [V] (dashed line) winds, and zonal mean precipitation [P], respectively. The 3rd column shows charts of precipitation P (color bar) along 950 hPa wind vectors. From top to bottom, rows show results for the climatological a) annual mean, b) boreal summer monsoon (JJA), c) austral summer monsoon (DJF), and d) the composite difference JJA – DJF, respectively.



Seasonal variations between JJA and DJF can be clearly seen in the zonal mean and geographically-varying structure of precipitation and horizontal wind vectors. Note there is a seasonal shift of the ITCZ from ~ 7°N in the annual and JJA averages to ~10°S in DJF (Fig. 2, 2nd column). In the composite difference JJA – DJF panel, [U] and [P] show roughly anti-
symmetric profiles with respect to the equator, whereas [V] is roughly symmetric, describing the annual cycle responses with a half-year time difference between the two hemispheres and the asymmetry of the Hadley circulation at the solstices.

The composite difference chart of precipitation and 950 hPa winds (Fig. 2d) highlights the prominent regional monsoon systems over the globe. During boreal summer monsoon JJA in the NH, heavy precipitation is associated with the North African, Asian, and North American monsoons (Fig.2b, golden shading in Fig. 2d). These monsoon precipitation
patterns are associated with dominant large-scale wind circulations. The northward flow in the boundary layer associated with the Hadley cell occurs in preferred longitude bands, including prominent cross-equatorial winds over Indonesia, over the western Indian Ocean (which feeds the Somali Jet), over the eastern Pacific, and eastern Atlantic Oceans, each of which brings moisture toward heavy precipitation regions along the ITCZ (Fig. 2 b, d). The North Atlantic and North Pacific are dominated by large surface anticyclones, which generate less precipitation but transport moisture clockwise from the subtropics, creating
a band of higher precipitation to the northwest of the anticyclones (Fig. 2b).

The monsoon circulation during austral summer DJF also exhibits a longitudinal preference for cross-equatorial flow in the boundary layer toward the summer hemisphere, notably in the western Pacific and western Atlantic, which feed into regional precipitation maxima over South Africa, the Amazon, and a band stretching from the South Pacific Convergence Zone (SPCZ) in the Southwest Pacific across the Indian Ocean (Fig. 2c). These regional precipitation maxima are highlighted by
green shading in the JJA – DJF difference plot (Fig. 2d). As in the NH during JJA, large anticyclones are prominent over the mid-latitude Indian, Pacific, and Atlantic Oceans of the SH during DJF, which help feed moisture poleward on their west edges, creating thin midlatitude precipitation maxima poleward of the anticyclones (Fig. 2c).

Seasonal characteristics of other meteorological variables are shown in Figure S2 of SI in (a-g) panels as JJA – DJF composite difference for (b) OLR, (c) upper tropospheric specific humidity, q, at 300 hPa, (d) SST, (e) MSLP with (U, V) at
950hPa, (f) horizontal wind divergence at 950 hPa, and (g) mid-tropospheric vertical velocity wind at 500 hPa. Seasonal variations in these meteorological variables exhibit systematic association with the composite difference of precipitation, P. Variations in OLR are somewhat smoother than those in P, having larger-scale patterns compared to the spatial scales of variation of the precipitation field. Deep convective features appear as opposite-sign peaks of OLR difference across all six regional monsoon systems. Seasonal changes in 300 hPa q and SST are nearly anti-symmetric about the equator. Seasonal
differences in MSLP are largest over eastern Eurasia (negative in JJA – DJF) and over the Pacific Ocean (positive in JJA – DJF) in the NH mid-latitudes. Horizontal divergence at 950 hPa shows air mass convergence into regions of higher precipitation. Small-scale features tend to dominate divergence fields, partly due to the effects of topography. The pattern of vertical wind at 500 hPa closely resemble the precipitation patterns, with stronger upward motion occurring over areas of higher precipitation. A more detailed description on the climatology and seasonality of all these meteorological variables can
be found in Yoden et al. (2023).





## 4. Zonal mean seasonal structure of QBO

In order to investigate the seasonal dependency of QBO dynamical teleconnections, the mean canonical form of zonal mean u-wind, and temperature is constructed between 60°S to 60°N in monthly de-seasonalized anomalies, separately for boreal and austral monsoon summer seasons. Figure 3 presents meridional sections of the composite canonical form of U' and T" for phase P4 (hereinafter QBO W 50 hPa) and phase P8 (hereinafter QBO E 50 hPa), along with composite differences P4 – P8 (hereinafter QBO W – E 50 hPa) for both JJA and DJF. During the boreal summer monsoon JJA, the U" composite difference does not reflect any statistically significant patterns in the troposphere on the NH side (Fig. 3c). However, on the SH or winter side, a noticeable pattern of QBO wind anomaly at 50 hPa over the equator extends poleward and downward in an arch shape into the subtropical troposphere, with penetration down to the surface. This arch is accompanied by a vertical anomaly strip of the opposite sign in the mid-latitudes, extending from the surface to the lower stratosphere (50 hPa). Along the UTLS, this dipole favors an equatorward shift of the zonal mean STJs during QBO W at 50 hPa.

The checkerboard patterns in T" anomalies (Figs. 3g, h, j, k) are consistent with the zonal wind anomaly patterns (Fig.s 3a, b, c, d) through the thermal wind relationship. This QBO temperature anomaly pattern is caused by the descending QBO MMC, which are more symmetric about the equator during the equinoxes and more asymmetric during the solstices (Hitchman et al., 2021). Stronger temperature anomaly patterns are seen in the winter subtropics (Figs. 3g, h, j, k). The subtropical QBO temperature anomalies in the lower stratosphere extend poleward and downward along the tropopause, especially on the winter side (SH during JJA and NH during DJF in Figs. 3g, h, j, k). This structure underlies the "horseshoe pattern" in zonal wind anomaly. The large QBO MMC anomalies which extend into the midlatitude stratosphere, causing the temperature anomalies (Fig. 3l), are consistent with the patterns of zonal wind response (Figs., 3 f) in the extratropical winter stratosphere, in the H-T mechanism and stratospheric route discussed by previous authors.

During DJF, the significant tropospheric temperature anomalies in the tropics (20°S-20°N) are consistent with a route along the subtropical troposphere (Fig. 3l). The QBO MMC in the UTLS simultaneously create antiphased temperature anomalies in the both tropics and subtropics. This implies that the QBO can affect the extratropics along the UTLS by either altering deep convection in the tropics directly, or modulating the temperature and zonal wind near the STJs. Since any "pure tropical forcing" would have to propagate through the subtropics to get to higher latitudes, the tropical-subtropical route is perhaps best thought of as one route along the subtropical UTLS. QBO anomalies in the tropical UTLS are directly linked with the surface region through modulation of deep convection, especially during DJF (Fig. 3l). QBO W at 50 hPa implies a warm anomaly near the tropical tropopause, which lowers the tropopause and stabilizes the upper troposphere, thereby reducing deep convection over Indonesia during NH winter (Collimore et al., 2003) and weakening the Walker Circulation (WC) (Yasunari 1990), and causing a warm anomaly throughout the troposphere over Indonesia (Muhsin et al., 2018). The opposite effect is found for QBO E at 50 hPa, with a cold anomaly in the tropical UTLS, warm anomalies in the subtropical UTLS in both the NH and SH, and enhanced convection over Indonesia during DJF. These results suggest that both the stratospheric and UTLS routes for QBO teleconnections operate simultaneously in addition to H-T mechanism during DJF, and confirm previous



studies showing an equatorially asymmetric, stronger extension of QBO MMC into the winter hemisphere (Randel et al., 1999;
Kinnersley, 1999; Pena-Ortiz et al., 2008; Hitchman et al., 2021; and Kumar et al. 2022).

**Figure 3.** Meridional sections of U" and T" anomalies for opposing QBO phase pairs during DJF and JJA. From left to right, columns show
anomalies for P4 (QBO W at 50 hPa) and P8 (QBO E at 50 hPa) in blue and red shading (color bar) and P4 − P8 (QBO W − E at 50 hPa)
in yellow and green shading (color bar). From top to bottom, rows represent U" wind anomalies for JJA (a–c), U" wind anomalies for DJF
(d–f), T" anomalies for JJA (g–j), and T" anomalies for DJF (j–l), respectively. The violet cross hatching indicates regions of statistical
significance less than 90%. The yellow and cyan dotted lines separate regions of fine and coarse contour intervals in the composite and
composite differences, respectively. Horizontal dashed lines indicate the 20 hPa, 50 hPa, and 100 hPa levels.



Figure 4 shows the same seasonal meridional structure of U″ and T″ composites as in Fig. 3 but for phase P5
(hereinafter QBO W at 70h Pa) and P1 (hereinafter QBO W at 70 hPa). Keying on the 70 hPa level generates a 9-element tic-
tac-toe pattern in the temperature field for both JJA and DJF (Figs. 4i, l) which is somewhat more coherent than keying on 50
hPa (Figs. 3i, l). During both JJA and DJF one may see the tri-pole pattern in temperature anomalies along the subtropical
UTLS (Fig.s 4g, h, j, k), with the subtropics antiphased with the tropics. This, in turn, creates QBO zonal wind anomalies
which also extend into the troposphere in a tri-pole meridional pattern, again with the tropical tropospheric zonal mean zonal
wind anomaly being opposite in sign to those in the subtropics (Figs. 4a-f). This subsequently modifies the strength and position
of the subtropical jets STJs. Note that moderate QBO anomalies in temperature and zonal wind are found throughout the
troposphere, which can affect surface weather.

**Figure 4.** Same as Fig. 3, but for P5 (QBO W at 70 hPa) and P1 (QBO E at 70 hPa).



The subtropical and extratropical tropospheric responses depend on season and phase of the QBO, with interesting differences seen even between using 50 hPa and 70 hPa as index levels (compare Figs. 3 and 4). For example, during DJF and QBO W at 70 hPa, a significant cooling anomaly is also evident in the lower troposphere poleward of 40°N (Fig. 4l). This confirms the recent finding of Kumar et al. (2024) regarding a new QBO teleconnection pathway for these cooling patterns via the UTLS region to the high-latitude surface which is independent of the H-T mechanism. Based on this zonal mean

analysis, QBO dynamical teleconnections with tropical, subtropical, and polar regions vary with the seasons and downward propagation the QBO anomalies. Additionally, the characteristics of QBO anomaly patterns in any latitudinal zone may differ with geographical location. Since the monsoon system is linked to the distribution of land and sea, it exhibits large geographical variations. Therefore, it is important to investigate the horizontal structure and variation of QBO teleconnections, therefore, from the next section onward, the analysis focuses on the non-zonal variations.

**5. QBO teleconnections with the GM system**

**5.1 Composite differences of seasonal precipitation and surface wind for opposite QBO phases**

        In order to visualize regional effects of QBO dynamical teleconnections in the GM system, we investigated the horizontal distribution of QBO anomalies in precipitation and surface wind for opposite phases of the QBO during the both JJA and DJF (Fig. 5). The index pairs P4 – P8 (50 hPa index) and P5 – P1 (70 hPa index) are of interest since they are associated

with significant QBO anomalies arriving in the tropical and subtropical UTLS. The top row represents the composite difference QBO W – E of precipitation anomalies, overlain with significant (U″, V″) difference arrows at 950 hPa for the 50 hPa and 70 hPa indices (i.e., P4 – P8 and P5 – P1) during JJA, and bottom row same for DJF. Significant patterns in the composite differences exist at continental scale in the equatorial, tropical, and extratropical regions which are related to modulation of regional atmospheric circulations for both seasons. As in the seasonal mean fields (Figs. 2b-c), one may observe a systematic

association between precipitation maxima and convergence in anomalous surface wind patterns. The differences among the four panels in Fig. 5 illustrate that regional monsoon system response to the QBO is sensitive to both seasons and QBO phase.

        During JJA and QBO W at 50 hPa (P4-P8, Fig. 5a), an east-west dipole in precipitation is seen in the tropical sector ~ 120°E - 200°E (Fig. 5a). There is enhanced precipitation over Indonesia. This is likely due to differences in the spatial pattern of deep convection (Fig. 2b, d). During JJA, deep convection shifts toward South East Asia, away from the Maritime Continent.

The South Asian High dominates the circulation in the UTLS. These differences in the basic state provide a different situation for the QBO MMC to interact, with modulation of convection occurring in different locations compared to DJF. The east-west dipole pattern modulates deep convection in the tropical west Pacific, which in turn affects the Pacific cell of the Walker Circulation (WC). QBO modulation of deep convection leads to modulation of the location and strength of lower tropospheric anticyclones (Fig. 2b), with QBO anomalies in the boundary layer circulation leading to changes in precipitation in subtropical

region. For example, there is more rainfall in the northwest Pacific during when there are QBO W at 50 hPa (Fig. 5a). This represents an eastward shift in precipitation, with a negative anomaly over China and southern Japan. Which is consistent with the results of Zhou et al. (2024), who showed that low precipitation preferentially occurs in the Yangtze –Huaihe River Basin

(YHRB) during QBO W 50 hPa. We will provide a detailed analysis of this point in a subsequent section. The composite difference QBO W– E at 70 hPa (P5-P1) during NH summer monsoon JJA does not show any significant alterations in
precipitation or wind vectors and only displays minor scale variations. (Fig. 5b).

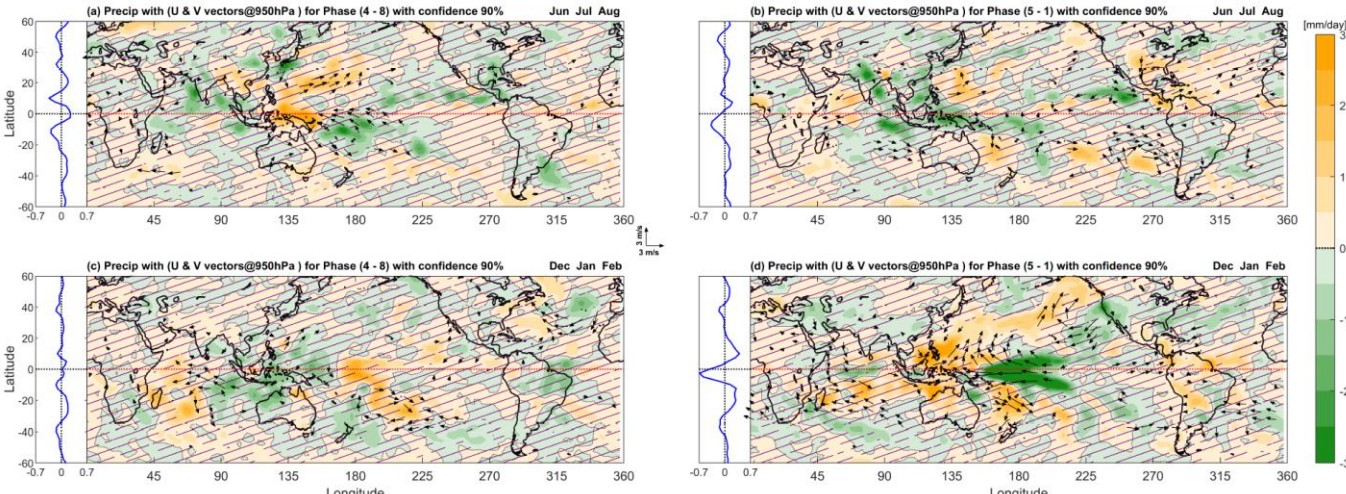

**Figure 5.** Longitude-latitude sections of QBO composite differences in precipitation (color bar) and 950 hPa horizontal winds (vector length shown in center inset) for QBO opposite phase pairs, for (a-b) JJA and (c-d) DJF. The left column represents QBO W – E at 50 hPa (P4 – P8), while the right column represents QBO W– E at 70 hPa (P5 – P1). The latitudinal profile of zonal mean precipitation is shown to the
left of each panel. Wind anomalies (U″, V″) are plotted if either component's statistical significance exceeds 90%. Violet cross-hatching indicates regions of statistical significance less than 90%.

Using the 50 hPa index (P4-P8), one obtains the opposite pattern in DJF (Fig. 5c) compared to JJA (Fig. 5a), the east-west dipole in the western tropical Pacific (~120°E - 200°E) switches sign, with less precipitation over Indonesia, in agreement with previous results (Collimore et al., 2003; Tegtmeier et al., 2020; and Hitchman et al., 2021). QBO W at 50 hPa implies
increased static stability in the tropical UTLS when there is a QBO warm anomaly near the tropopause (Gray et al., 1992b; Giorgetta et al., 1999). Consistent with this, Liess and Geller (2012) report a diminution of rainfall over Indonesia in DJF during QBO W. The reduced precipitation over Indonesia (Fig. 5c) is consistent with our zonal seasonal mean structure (Fig. 3l) and non-zonal profile of tropopause structure shown later (Fig.6c), since that implies the existence of a warm anomaly near the tropical tropopause over the deep convective region of Indonesia. The east-west dipole in the western tropical Pacific
during DJF also switches sign when the pair P5 – P1 (Fig. 5d) is used instead of P4 – P8 (Fig. 5c). Along the equatorial region, the patterns resemble those of JJA for 50 hPa, exhibiting significant modulation in the Pacific cell of the WC and a dipole pattern in precipitation. However, the DJF patterns are broader with higher amplitudes. DJF also led to significant modulation in the extratropics, for 50hPa index, a significant modulation of precipitation (an east-west dipole) and wind vectors (semicircular anticyclonic circulation) can be seen over the North Atlantic Ocean (Fig. 5c), where the localized atmospheric
circulation known as the Northern Atlantic Oscillation (NAO) prevails. For the 70 hPa index, the significant semicircular anticyclonic circulation and dipole pattern in precipitation over the Pacific–North American (PNA) region further highlights that QBO dynamical teleconnections occurs in regions where localized atmospheric circulations are present.



To explore how QBO anomalies behave in a non-zonal pattern near the tropopause and its possible teleconnection with GM system, figure 6 shows composite QBO differences similar to Fig. 5, but for 100 hPa temperature and 100 hPa

horizontal winds. During JJA, anomalous QBO westerly winds (range ~ 2-3 m/s) and a warm anomaly (range ~ 1-3 K) are found near the tropopause in a zonally-symmetric pattern along the equator. This is true for both the 50 hPa index level (Fig. 6a) and 70 hPa (Fig. 6b), but the warm anomaly is more prominent for the 70 hPa index. Note also the prominent QBO cold anomaly near 20°S during JJA, as shown by Hitchman et al. (2021), it is associated with the same MMC which caused the warm anomaly over the equator, resulting in a stronger STJs in the SH winter (Figs. 6a, b). The subtropical winter cooling

effect along the UTLS generated by the QBO MMC is stronger and more significant for the 50 hPa index (Fig. 6a). In agreement with the zonal mean structure during JJA, significant QBO negative temperature anomalies in the subtropics associated with the QBO MMC have zonally symmetric features in the winter hemisphere (Figs. 6a, b).

In the NH summer JJA, the 50 hPa composite difference reveals a distinct cool anomaly over the subtropical western Pacific Ocean near 140°E (Fig. 6a) which extends into the Bering Sea and coincides with the region where a significant pattern

was found in precipitation and wind vectors at the surface (Fig. 5a). The composite difference for the 70 hPa index also shows a region of cooling in the subtropical lower stratosphere over the central Pacific midlatitudes (Fig. 6b), although less robust than for the 50 hPa index. These results indicate that there are preferred longitude bands where the QBO can influence the extratropics along the subtropical UTLS, with subsequent modulation of extratropical surface weather.

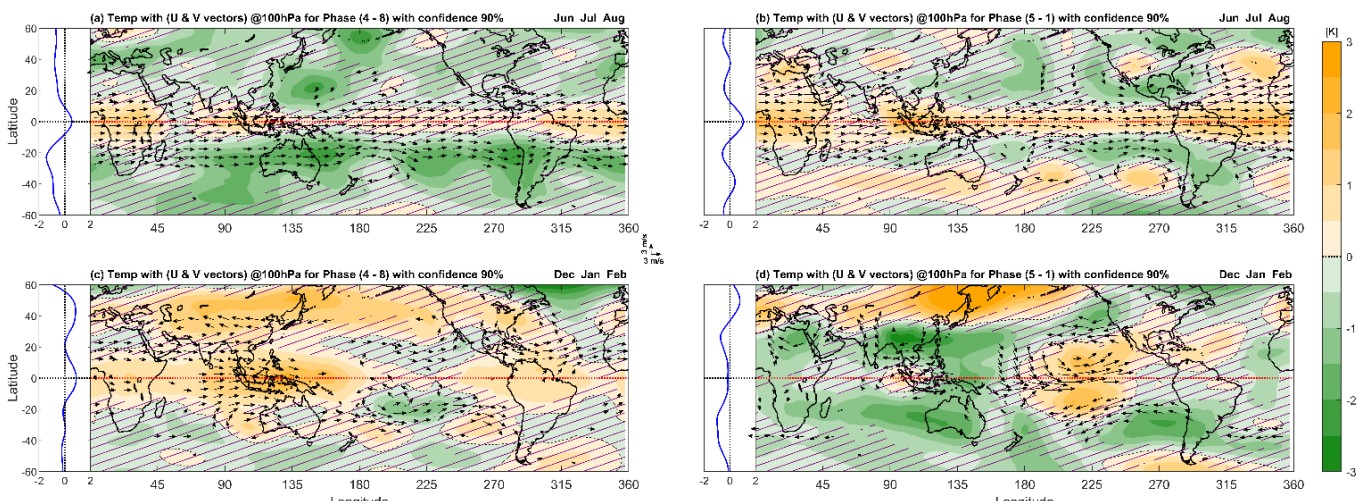

**Figure 6.** Same as Figure 5 but for temperature (T", color bar) and wind vectors (U", V") at 100 hPa. The zonal mean meridional profile of QBO temperature anomaly is shown to the left of each panel.

During DJF, QBO westerly wind and warm temperature anomalies appear in the tropics at specific longitude bands (Figs. 6 c, d). For the 50h Pa index, a significant QBO warm anomaly occurs over Indonesia, extending from central Africa to the Date Line, with another warm anomaly over Amazonia (Fig. 6c). The range reaches 3K over Indonesia where significant

westerly wind vectors reside. These warm anomalies coincide with the locations of chronic deep convection, characterized by



cloud top temperatures less than 192 K and low OLR emission (Collimore et al., 1998) and an easterly flow anomaly at 150 hPa, just below the tropical tropopause, consistent with a reduced WC during QBO W (cf. Figs. 18e, 19e of Hitchman et al., 2021). Similar temperature anomaly patterns were observed by Hitchman et al. (2021) in cold point tropopause temperature (CPT) for the same QBO W – E phases (see their Fig. 17b). Tropical UTLS thermal anomalies associated with the monsoon

system are stronger during DJF, with greater geographical variation in the strength of deep convection and tropopause height. This creates a greater opportunity for QBO anomalies to preferentially affect regions with characteristically deep convection during DJF compared to JJA.

Significant QBO westerly wind anomalies can be seen in the NH and SH subtropical eastern Pacific and in the North Atlantic (Fig.6c), which was noted as downward-extending subtropical anomalies in the zonal mean structure (Figs. 3f).

Extratropical QBO warm anomalies appear near 50°N and 30°S, being stronger in the winter hemisphere and, in the eastern hemisphere (Fig. 6c). A cold anomaly can be seen at 100 hPa centered near the tip of Greenland, consistent with an amplified wave-one pattern during QBO W (Kumar et al., 2024). This North Atlantic 100 hPa temperature anomaly is related to the westward shift in the precipitation maximum (Fig. 5c). This geographical fingerprint provides useful information for evaluating whether this teleconnection operates through the stratosphere pathway in the H-T mechanism or along the subtropical UTLS.

Composite differences in 100 hPa temperature for QBO W – E at 70 hPa level shows a dipole pattern in the tropical Pacific and PNA regions, as was observed for the surface precipitation (Fig. 5 d). A wavenumber-1 pattern in temperature anomaly is also evident near 60°N (Fig. 6d), similar to that observed at 50 hPa (Fig. 6c). There are also significant easterly anomalies in the STJs of both hemispheres in the eastern Pacific when the 70 hPa level is used, resulting in a cyclonic circulation off of the west coast of North America (upper level trough), which is reverse to anticyclonic surface wind vector anomaly (Fig. 5d).

These composite differences maps illustrate the systematic, significant modulations of the GM system by OBO anomalies in the UTLS, precipitation, and surface winds throughout the tropics and extratropics. The characteristic of this dynamical teleconnection is regional and is manifested in regions where prominent localized atmospheric circulations prevail. Results are sensitive to QBO phase. In the tropics a QBO signal in precipitation is observed, which occurs in centers of deep convection. The strongest signals are associated with modulation of the Pacific component of the WC and exhibit significant

dependence on season and on QBO index level. The zonal mean QBO MMC causes simultaneous antiphased cold and warm anomalies in the tropical and subtropical UTLS. In the tropics the QBO modulates regions of deep convection and therefore creates a regionally-varying signal which can also propagate into the extratropics along the UTLS. The manner in which the teleconnection signal propagates from the tropics into the midlatitudes is not fully understood but involves modulation of planetary Rossby wave radiation from tropical convective centers. Locally this can lead to changes in seasonal mean cyclonic

and anticyclonic circulations in the extratropics, thereby leading to enhanced or diminished local meridional overturning in "local Hadley cells". During the NH summer the stratospheric route is disabled. Therefore, during summer, the QBO primarily influences the extratropics via the subtropical UTLS pathway, but during winter a QBO teleconnection can reach the extratropical surface via either the stratospheric pathway and annular modes or via the subtropical UTLS. The rest of this study will investigate precise information on these potential routes with their regional characteristics.





## 5.2 QBO dynamical teleconnection with the northwest Pacific during JJA


In order to investigate dynamical teleconnections operating along the subtropical route during boreal summer, we analyze composite climatologies of precipitation and other meteorological parameters, for QBO phases P4 and P8, along with composite differences. The northwest Pacific region (10°-50°N, 120°-190°E) is shown in Fig. 7 during JJA for the 50 hPa QBO index (P4 and P8). From top to bottom, rows show precipitation, 500 hPa vertical wind (W), MLSP together with 950 

hPa horizontal winds, and 100 hPa temperature together with 100 hPa horizontal winds. The presence of the QBO signal is clearly visible even in the composite climatologies for P4 and P8. During QBO E at 50 hPa, the band of precipitation which arches anticyclonically toward Alaska is enhanced south of Japan (Fig. 7b), with associated increase in mid-tropospheric upward motion (Fig. 7e). During QBO W at 50 hPa there is less precipitation along this summertime synoptic-scale storm track, more precipitation throughout the rest of the northwest Pacific (Fig. 7a), and reduced upward motion near the southern 

tip of Japan (Fig. 7d). These features are more easily seen in the difference plots (P4 – P8, Figs. 7c, f).

This variation in precipitation is associated with modulation of the regional circulation (Figs. 7g-l). During QBO W at 50 hPa, the lower tropospheric anticyclonic which dominates the northwest Pacific basin is diminished (Fig. 7g), while it is enhanced during QBO E at 50 hPa (Fig. 7h), such that a negative anomaly in MSLP and cyclonic anomalous circulation is seen in the difference composite QBO W – E (Fig. 7i). During the QBO W phase the high-pressure region contracts, with 

isobars shifting slightly eastward (as indicated by the position of the yellow line in Fig. 7g). This shrinking of the high-pressure zone is associated with diminished northward flow towards Japan. The resultant change in the anticyclone circulation will bring less precipitation south of Japan and along the periphery of the anticyclone, with enhanced precipitation in this region during QBO E. Heavy precipitation is associated with rising air currents; therefore, the vertical wind should be stronger over such areas. These upward winds do not remain limited within planetary boundary layers and penetrate beyond the mid-

troposphere into the upper troposphere.

At 100 hPa, it can be seen that the northwest Pacific region during JJA is dominated by the northeastern portion of the South Asian High, which dominates the lower stratospheric circulation in the subtropics from the western Pacific westward to the Atlantic Ocean (Figs. 7j, k, l). Note the strong anticyclonic curvature of the flow near 30°N, 120°E. During QBO W at 50 hPa this circulation is stronger, with an associated cold anomaly at 100 hPa (Fig. 7j), while during QBO E this circulation 

is weaker and 100 hPa temperatures are not as cold (Fig. 7k). This results in composite difference fields such that the high cold anticyclonic dome extends upward further into the stratosphere during QBO W, with enhanced southward flow in the central North Pacific on the eastern edge of the QBO anomaly (Fig. 7l).





**Figure 7.** QBO regional teleconnnection manifestation in the Northwest Pacific (10°-50°N, 120°-190°E) during boreal summer (JJA), showing composite climatolgies for QBO phases (left) P4, (middle) P8, and (right) P4-P8, for (a-c) precipitation, (d-f) mid tropospheric W wind at 500 hPa, (g-i) MLSP with (U, V) wind at 950 hPa, and (j-l) 100 hPa temperature with (U, V) wind. The violet cross hatching on the composite difference plots indicates regions of statistical significance less than 90%. Wind anomalies (U″, V″) are plotted only if either component's statistical significance exceeds 90%.

As shown in Fig. 6a, a significant QBO MMC signal was found over this region only, and composite differences in 100 hPa temperature anomalies show similar significant patterns as MSLP, but on a larger scale (Fig. 7 l). The South Asian High appears to be more robust in its northeast quadrant during QBO W at 50 hPa. This analysis suggests that, in the subtropical route, the QBO can connect to surface dynamics at a regional scale via modulation of localized atmospheric circulations.

## 5.3 QBO teleconnections with the North Atlantic during DJF for 50 hPa index

Figure 8 is in the same format as Fig. 7, but focuses on the North Atlantic region (20°-60°N, 280°-350°E) during austral summer. During DJF, teleconnection pathways are possible, via either the stratosphere, or along the UTLS, or both. For QBO W at 50 hPa (Fig. 8a), precipitation is enhanced in the western North Atlantic and diminished in the eastern North



Atlantic relative to QBO E (Fig. 8b). This feature is distinctly recognizable as a significant east-west dipole in the composite difference QBO W – E (Fig. 8c), indicating westward shift in precipitation during QBO W. A similar east–west dipole pattern is also evident in the anomalous vertical motion at 500 hPa, reflecting a westward shift in upward motion during QBO W (Fig. 8 f). The variation in precipitation is closely linked to changes in the GM system circulation patterns (Figs. 8g-l). During QBO W at 50 hPa, the dominant Azores High surface anticyclone is strengthened (Fig. 8 g), with a westward shift in poleward advection of moist tropical air. This signal is quasi-barotropic, extending from the surface to the lowest stratosphere at 200 hPa (Fig. 8l). It represents a strengthening of anticyclonic flow in the North Atlantic during QBO W at 50 hPa. A stronger Azores High implies a higher index NAO and stronger northeastward flow toward Scandinavia in the northern North Atlantic (e.g., Thompson and Wallace 2000).



**Figure 8.** As in Fig. 7, but for DJF and focused over the North Atlantic Ocean (20°-60°N, 280°-350°E).

The increased pressure in the Azores High enhances the anticyclonic circulation near the Azores, resulting in a more intense North Atlantic (NA) jet stream. This change in anticyclone creates optimal conditions for heavy precipitation in the western NA including the eastern coast of North America and lighter precipitation around the Azores High (Fig. 8a). The




opposite scenario is found for QBO E at 50 hPa (Fig. 8b). The composite difference QBO W − E clearly highlights this feature, showing a significant anticyclonic motion in wind anomalies centred around the high-pressure zone (Fig. 8i). Similar to JJA, stronger upward motion can be observed at 500 hPa (Fig. 8d to f) over regions of heavy precipitation.

In the extratropical lower stratosphere (200 hPa), temperature anomalies display significant patterns akin to those of MSLP (Figs. 8i, l). This is contrast to the situation in the northwest Pacific during JJA, where a large surface anticyclone is centered over the Pacific and a larger anticyclone is centered over Tibet in the UTLS (Figs. 7i, l). In the North Atlantic during DJF the structure is equivalent barotropic, with a strengthened anticyclone extending upward, causing cooling of the lower stratosphere in the eastern North Atlantic during QBO W at 50 hPa (Fig. 8i, l). A similar relationship is seen between the
strengthened northeastward extension of the South Asian High and the cold anomaly at 100 hPa during QBO W in JJA (Fig. 7). As discussed earlier in the seasonal DJF zonal mean structure for the 50 hPa index, these anomaly patterns may be linked to either the H-T mechanism associated with the stratospheric QBO teleconnection route, or to the route along the subtropical UTLS, or to a combination of both. But, again refining that the influence of the QBO is manifested at a regional scale where localized atmospheric circulations exist. In this case, the QBO is regionally interacting with the NAO.

**5.4 QBO teleconnections with the northeast Pacific during DJF for 70 hPa index**

Figure 9 shows the same variables as in Fig. 8, but for the northeast Pacific region (20°-60°N, 190°-260°E) and using the 70 hPa index. QBO interacts with the PNA pattern for 70 hPa index. During QBO W, rainfall is enhanced with along 500 hPa vertical motion (Fig. 9 a, d) from northwest of Hawaii to the Gulf of Alaska, while these are diminished along the west coast of the United States. The reverse scenario during can be seen during the QBO E phase.

Consistent with previous results (section 5.2 & 5.4), these patterns are systematically linked with variations in the localized atmospheric circulations. QBO W at 70 hPa modulates the tropospheric planetary wave pattern in this region such that a positive PNA pattern is excited, with an enhanced surface low in the Gulf of Alaska and enhanced anticyclonic flow off the west coast of California (Fig. 9g). A stronger tropospheric low in the Gulf of Alaska implies a lower tropopause, hence a warm anomaly in the lower stratosphere near 200 hPa (Fig. 9l). The effect of the QBO on the PNA is similar to that of ENSO,
where QBO W at 70 hPa or La Nina favors more precipitation in Alaska, while QBO E at 70 hPa or El Nino favors more precipitation in California during DJF. Note that our analysis includes only the neutral ENSO phase, and obtained the same patterns even with a low threshold of the ENSO index (±0.5 K). Therefore, any contamination from ENSO events can be ruled out.



**Figure 9.** As in Fig. 7, but focused over the Northeast Pacific region (20°-60°N, 190°-260°E) during DJF, for QBO W and E at 70 hPa (P5 and P1).

## 6. Longitude-altitude structure of regional QBO anomalies

The above analysis suggests that the QBO has dynamical teleconnections with different parts of the GM system via modulation of regional atmospheric circulations. The anomalies in MSLP, surface winds, and precipitation patterns were found to be associated with QBO footprint in the UTLS. To better understand the vertical structure of perturbations associated with QBO, we analyze longitude-altitude sections of anomalies in geopotential height (GPH), temperature, static stability, meridional wind, and vertical wind. These analyses are averaged over selected latitude bands for the opposite QBO phases at which significant composite difference patterns were observed for three regions: the northwest Pacific during JJA, the North Atlantic Ocean during DJF, and the northeast Pacific during DJF (Fig. 10).




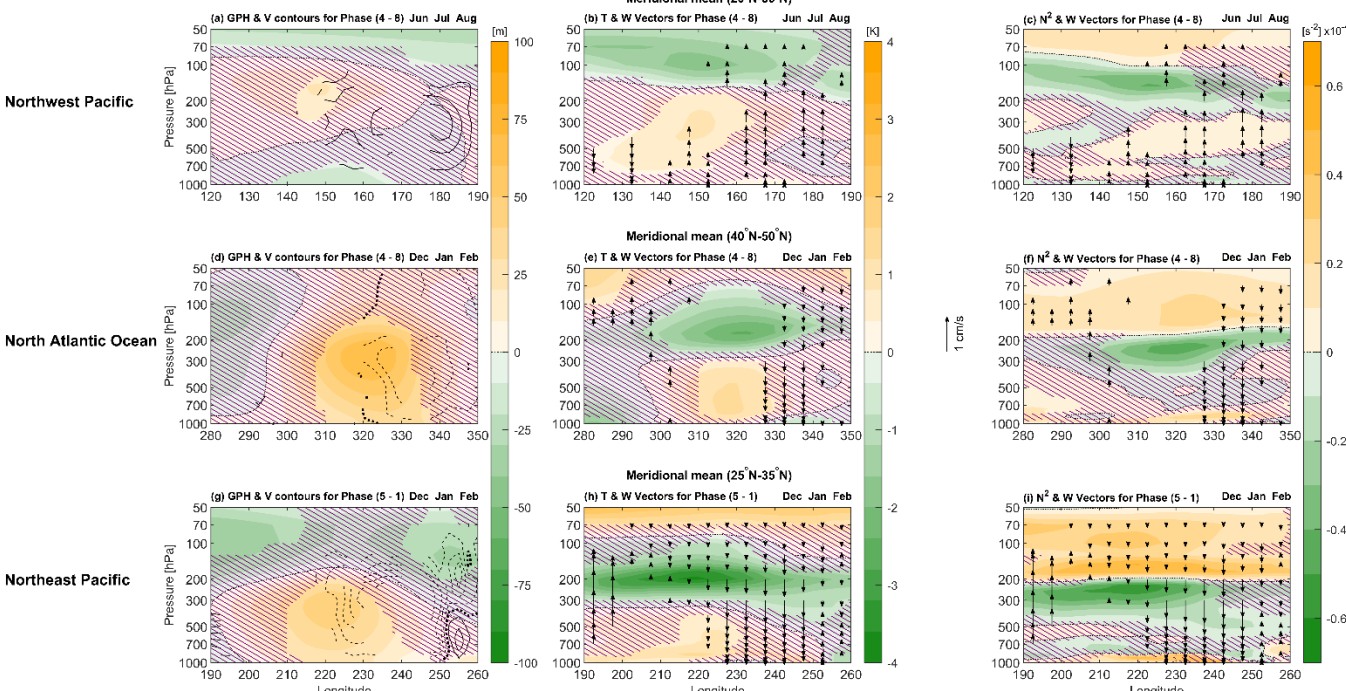

**Figure 10.** Longitude-pressure sections of composite differences between opposite QBO phases for selected latitudinal means shown in Fig. 7, 8, and 9.  The top row shows the latitudinal mean between 20°N and 30°N over the Northwest Pacific (120°E - 190°E) during JJA, the middle row shows the mean between  40°N and 50°N over the North Atlantic Ocean (280°E – 350°E) during DJF, and bottom row shows the mean between 25°N and 35°N over Eastern Pacific (190°W and 260°W) during DJF. The first, second, and third column of each row represent the composite difference of GPH″ (color bar) and V″ wind (contour lines), T″ (color bar) with W″ wind vectors, and N″ $^2$ with W″ wind vectors, respectively. The top two rows use QBO W – E at 50 hPa (P4 – P8), and the third uses 70 hPa indices (P5 – P1). 5NViolet cross-hatching indicates regions of statistical significance less than 90%. The V″ contours on the 1$^{st}$ column and W″ wind arrows on the 2$^{nd}$ and 3$^{rd}$ column are plotted only if their statistical significance exceeds 90%. I

In the northwest Pacific during JJA, composite difference in GPH  anomaly shows a diminution of the surface anticyclone (Fig. 10a) and cold anomaly in the lower stratosphere (Fig. 10b). The stratospheric low GPH anomaly is associated with the cool anomaly. Interestingly, the tropospheric warming patterns are limited in longitudinal range where significant modulation was observed in the MSLP. The static stability (Brunt–väisälä frequency) patterns shows a statistically significant alternating tripole patterns: positive in stratospheric region (above 70 hPa), negative around the tropopause (200 hPa -100 hPa), and positive in the mid tropospheric region (700 hPa – 300 hPa) (Fig. 10c). In the upper troposphere, the enhanced northeastward extension of the South Asian High during QBO W can observed as a high GPH anomaly (Fig. 10a), with reduced static stability above (Fig. 10c). Significant upward motion anomalies at the upper troposphere can also be seen (Fig. 10c), with significant positive v-wind contours representing the meridional circulations on the northward side (Fig.10a). These systematic modulations from tropospheric to stratospheric region, as well as the meridional linkage, suggest the possible role of the QBO in modulating precipitation and other meteorological variables in regional circulation systems.



Zhou at el. (2024) analysed the QBO meridional structure averaged over 120°E-160°E for the summer monsoon, JJA, and showed that during QBO W at 50 hPa there are regional zonal wind anomalies in the troposphere in the band 30°N-40°N, This corresponds to an equatorward shift in the STJs during QBO W, which coincides with reduced moisture transport and convergence, leading to lower rainfall extremes in the YHRB. Gao et al. (2023) described these results in terms of a regional QBO MMC anomaly. Our results confirm low precipitation over this region for QBO W at 50 hPa, but these features are not

limited to the YHRB, having broader spatial domains spanning the northwest Pacific.

In our analysis, we found a regional cool anomaly in the upper troposphere over the northwest Pacific during QBO W at 50hPa. This is related to enhanced deep convection in the Southeast Asian monsoon, the northeastward expansion of the South Asian High in the upper troposphere, and diminution of the surface anticyclone over the northwest Pacific. These changes result in reduced upward motion and precipitation over the southern tip of Japan and enhanced upward motion and

precipitation over the northwest Pacific. It is possible to call this a local MMC anomaly, where the polarity is rising over the northwestern Pacific and sinking over Japan during QBO W at 50 hPa, but the underlying cause for this apparent local circulation is more complex.

Considering the zonal mean temperature anomalies in Fig. 3i, one may see a QBO cold anomaly all along the subtropical UTLS during QBO W at 50 hPa, which accompanies the warm anomaly over the equator. The zonal mean QBO

cold anomaly in the subtropics is caused by a rising branch of the QBO MMC, while the warm anomaly is caused by the sinking branch in the tropical UTLS. This subtropical cold anomaly might be expected to enhance precipitation in a zonally-symmetric pattern, since there is a decrease in static stability in the subtropical UTLS across Asia and the northwest Pacific, from 120°E to 180°E (Fig. 10c). This would encourage more deep convection over the northwest Pacific relative to over land along the east coast of Asia during QBO W. Another aspect of this system which may be important in this context is that deep

tropical convection is suppressed over Indonesia during QBO W, which implies reduced subtropical downwelling. The lower tropospheric anticyclone which dominates the circulation east of Japan during summer (the Bonin High, Enomoto et al., 2003) is one region of subsidence in such a local Hadley Cell. With reduced convection in the adjacent tropics, one might expect a reduction in subsidence and strength of the Bonin High, which is observed (Figs. 7f, i and 10a).

To see other possible routes for QBO dynamical teleconnection, potentially through the polar or extratropical region,

orthographic NH polar projections (0°N–90°N) of GPH" at 50 hPa, along with 200 hPa tropospheric temperature, MSLP, and precipitation, are shown in Fig. S3 for JJA. None of the panels exhibit any other possible routes for QBO teleconnections at the 50 hPa level (2nd column, Fig. S3) or even at 70 hPa (3rd column, Fig. S3), because quasi-stationary planetary wave activity between the equator and polar regions remains very low during the summer monsoon season when stratospheric winds are easterly (Chen et al., 2005). The orthographic polar projections also highlight the localized characteristics of the QBO MMC

subtropical UTLS route over the northwest Pacific, as previously discussed.

During DJF, the nature of dynamical teleconnection remains intact with the downward propagation of the QBO phase (50hPa to 70hPa), but the geographical location for surface linkage changes. For both the regions, North Atlantic Ocean during DJF and northeast Pacific, the significant positive GPH anomalies prevails in the entire tropospheric region but within



respective longitudinal range where significant composite difference was found in the MSLP (Fig. 10d and g). The Atlantic patterns are vertically deeper in compare to Pacific. Interestingly, temperature anomalies of both regions reflect the significant dipole pattern; positive anomalies in the tropospheric region below 300hPa and negative in upper tropospheric region from 300hPa to 100hPa (Fig. 10 e and h). The static stability also exhibits a significant dipole pattern centered around the peak amplitude of negative anomalies in the temperature dipole pattern ~200 hPa.

To better understand QBO teleconnections during DJF, we also analyzed the horizontal distribution of various dynamical parameters on orthographic NH polar projections (0°N–90°N), which are shown in Fig. 11. Over the Arctic Circle (>60°N), significant composite differences, QBO W – E, seen in GPH″ at 50 hPa are indicative of a deeper polar vortex during QBO W than QBO E (2nd column Fig, 10 a). This annular mode structure is part of the teleconnection pathway originally known as the QBO stratospheric route, or H-T Mechanism. The lower stratospheric T″ anomalies at 200 hPa are somewhat similar to those at 50 hPa GPH″ (2nd column Fig. 10b), consistent with a deeper and cooler polar vortex. However, MSLP″ patterns are rather independent from GPH″ (2nd column Fig. 10c), showing the association with NAO as discussed in section 5.2. The NAO and polar vortex are linked by annular mode structures (Kodera et al., 1999, Kumar et al., 2022). The QBO-NAO teleconnection may result from QBO-induced changes in the tropospheric circulation anomalies via the downward control principle (Haynes et al., 1991), and zonal flow–stationary wave interactions. Another possible mechanism for this teleconnection may be associated with QBO latitudinal displacement of the STJs. The relationship between zonal mean QBO anomalies associated with the QBO MMCs and regional circulations is complex, with distinctly different responses of the local STJs over the Pacific and Atlantic and during DJF and JJA (Kumar et al., 2022). However, a detailed mechanism for the NAO-QBO dynamical teleconnection is still unclear and needs to be explored in the future.

For the 70 hPa index pair P5 – P1, the location of statistically significant QBO modulation shifts from the North Atlantic to the northeast Pacific, but the characteristic nature of vertical coupling is found to be similar in both the regions (compare 2nd and 3rd rows of Fig. 10). In the longitude-altitude structure, the significant negative v-wind contours representing the meridional circulations on the southward side with downward wind (Fig.10g, h). The composite difference QBO W– E (P5–P1) of GPH″ does not reflect any annular mode structure, i.e. no H-T mechanism. Instead, a wave-number-one-patterns exists (3rd column in Fig. 11a), with an amplified Aleutian High over the Pacific and trough over the North Atlantic (Harvey and Hitchman 1996) during QBO W at 70 hPa and DJF. Although the trough anomaly is not highly significant, the Aleutian High anomaly is. The lower stratospheric temperature patterns at 200 hPa are consistent with GPH″, where a warm anomaly underlies the Aleutian amplified anomaly over the North Pacific and a cold anomaly underlies the trough over the North Atlantic. This ridge-trough pair tilts westward with height from the troposphere into the upper stratosphere and mesosphere (Harvey and Hitchman 1996). Thus, the QBO can modulate this fundamental structure. Interestingly, a significant MLSP anomaly lies in Article Circle as a semi-disc structure (0°E–180°E), along with some significant patterns in the northeastern Pacific region which modulates the PNA as discussed above.





**Figure 11.** NH orthographic polar projections (10°N–90°N) of a) 50 hPa GPH, b) T at 200 hPa, c) MSLP, and d) precipitation, showing the climatology and anomalies of composite differences between QBO opposite phases during DJF and neutral ENSO. The first, second, and third, columns of each row represent the climatological mean, QBO W – E at 50 hPa (P4 – P8), and QBO W – E at 70 hPa (P5 – P1), respectively. Black dots on the both composite difference plots highlight regions where statistical significance exceeds 90%.

One of the possible mechanisms for the QBO – PNA teleconnection is modulation of the planetary wave train emanating from tropical convection centers along the UTLS pathway (Kumar et al., 2024). They showed that enhanced EP flux convergence in the high latitudes during QBO W at 70hPa. Increased extratropical stratospheric EP flux convergence favours a low index annual mode for both the SH and NH (Polvani et al., 2010, Fig. 6 of Kumar et al., 2024). Our results also





support the idea that the UTLS pathways involves modulation of the annual ar mode, whose primary response is a north -south displacement of the midlatitude jet in a dipole pattern (cf. Plate 4 in Kushner, 2001 and Fig. 4, Kumar et al., 2024).

## 7. Conclusions

This study presented a brief overview of GM system as documented in the 42-year climatology, using global monthly
mean ERA-5 reanalysis data in the satellite era from 1979 to 2022. We focussed on possible teleconnections between the stratospheric QBO and different monsoon systems at the global scale, in the absence of extreme El Niño and La Niña events. We explored modulation of regional circulations during two different seasons - boreal summer JJA, and austral summer DJF. Composite differences were emphasized for whenever QBO anomaly lies in the lower stratosphere, at 50hPa and 70 hPa. To obtain better insight into QBO pathways for teleconnections with the GM, the seasonal mean canonical form of the QBO
anomalies was investigated separately for boreal summer monsoon JJA and austral summer monsoon DJF. Teleconnections were found to vary with the seasons, and to be sensitive to QBO phase. Previous findings regarding the equatorial QBO routes were confirmed: the tropical-subtropical route along the UTLS dominates during the boreal summer JJA. Analysis suggested that the QBO has dynamical teleconnections with different regional monsoons where prominent dynamical circulations prevail, and primarily modulates these circulations, thereby influencing the associated precipitation patterns. Significant modulations
of precipitation patterns were observed across the three main regions of study, the northwest Pacific during JJA, and the North Atlantic and northeast Pacific during DJF.

During JJA, the QBO influences tropical precipitation by modulating the WC. QBO W at 50 hPa intensifies the Pacific cell of WC (120°E to 225°E), which brings heavier rainfall over the Maritime Continent and less over the western Pacific. This pattern can be attributed in the modulation of convection, driven by QBO-generated thermal anomalies in the
UTLS. In addition, a route along the subtropical UTLS was observed as modulation of circulations over the northwest Pacific which impacted the associated precipitation in a vast region extending from China to the western Pacific. It is confirmed that QBO W 50 hPa brings lower precipitation over YHRB region (Zhou at el. 2024). These features were not confined to this region alone, rather, they extended across broader spatial domains, from Japan to the northwestern Pacific Ocean. QBO W at 50 hPa favors a shift in rainfall from the tip of Japan eastward toward a diminished Bonin high. The opposite scenario was
observed during QBO E. These patterns can be understood in terms of a regional response to zonally symmetric temperature anomalies associated with the QBO MMC in the subtropical region that prevailed from northeast Asia to north Pacific region, together with diminution of the Lagrangian overturning circulation between reduced deep convection over the Maritime Continent and a reduced equivalent-barotropic Bonin High.

During DJF, the presence of QBO teleconnection routes along the subtropical UTLS and stratospheric-polar routes
were evident through the modulation of precipitation in the extratropics. In the North Atlantic region, QBO W at 50 hPa favors a positive NAO index. This, in turn, enhances the anticyclonic circulation associated with the Azores High, resulting in a more intense NA jet stream and shifts the precipitation maximum westward toward the east coast of North America away from around the Azores High. QBO W at 70 hPa promotes a positive PNA phase. As a result, strengthens the anticyclonic circulations over northeast Pacific and intensifies the mid-latitude north-eastward flow over the Pacific Ocean, which brings



above average precipitation to southern Alaska and below average across the west coast of the United States. The opposite scenarios occur for QBO E. On the basis of the current analysis, it is difficult to discern the exact mechanism for these linkages, as most analysis focused on phenomenological descriptions. However, the QBO-NAO linkage may be understood in term of the H-T mechanism associated with the QBO polar route. The NAO and polar vortex have stronger dynamical teleconnection with more annular mode structure (Kodera et al., 1999, Kumar et al., 2022). A deeper polar vortex was observed for QBO W

than QBO E, affecting the annular mode structure. Another possible mechanism for this linkage may be QBO displacement of the STJs. The QBO-PNA linkage may be due to changes in the pattern of planetary wave radiation along great circle routes from seasonal monsoon system convective centers via the UTLS pathway, with enhanced EP flux convergence in the high latitudes (Kumar et al., 2024). Further detailed investigations are necessary to understand the precise mechanisms operating in the QBO-NAO and QBO-PNA teleconnections.


**Data availability statement:** All data used in this study are openly accessible for the public.

The ERA-5 data set is available online at https://cds.climate.copernicus.eu/datasets

The OLR data is available online at https://psl.noaa.gov/data/gridded/data.olrcdr.interp.html

The GPCC data is available online at https://psl.noaa.gov/data/gridded/data.gpcc.html

The HadISST data set is available online at https://www.metoffice.gov.uk/hadobs/hadisst/data/download.html

**Acknowledgments:** The authors would like to thank all members of the ERA5 reanalysis, HadISST, GPCC, and NOAA teams for their efforts in making these datasets available online.

**Competing Interests:** The authors declare that there are no competing interests.

**Research Funding:** This paper is based on achievements of the collaborative research project (2025IG-04) of the Disaster Prevention Research Institute of Kyoto University.

**Author Contributions:** VK and SY jointly worked on the research design and methodology. VK analyzed the all data and wrote the first draft of the paper. MHH, SY, TT, and KI contributed to the conceptualization, review, and edited the draft. MHH also contributed significantly to the discussion section.




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
