# Peer review of "Quasi-Biennial Oscillation Modulation of Global Monsoon Systems and Regional Teleconnections"

_EGUsphere, 2025_

## Referee Comment (RC1)

**Review of manuscript #2025JD03420. First Round**

September 19, 2025

This manuscript examines the influence of the Quasi-Biennial Oscillation (QBO) on global monsoon rainfall and related teleconnections using ERA5 and GPCP data. The authors emphasize seasonal differences, vertical structure (50 vs. 70 hPa winds), and the apparent linearity of phase progression. While the topic is timely and of interest, the manuscript in its current form requires substantial revision. The main weaknesses are an incomplete and selective literature review, a tendency to assert rather than demonstrate mechanisms, and an relatively simple, and my opinion therefore unconvincing, treatment of ENSO. I recommend major revisions.

**Major Comments**

1. **Incomplete and selective literature review**

   - The Introduction omits several key works on monsoons as well as QBO impacts on convection and precipitation.

     The authors frequently argue that the global monsoon is a land-sea contrast phenomena, overlooking recent literature on the monsoons as a results of energetics and solar forcing (Bordoni and Schneider, 2008; Biasutti et al., 2018; Geen et al., 2020), which include papers they cite on their manuscript (Wang and Ding, 2008). If the authors choose to delineate the global monsoon as the ocean-land thermal contrast only, they should at least mention other existing views and explain why they choose this approach. I realize this is only tangentially related to the whole manuscript but it's an important distinction.

     Then, the authors provide only a handful of references and insight into the tropical connection between the QBO and the surface. For example, they cite Yasunari correctly as a paper that relates the QBO to the Walker circulation strength but they fail to also mention more recent work that has found similar evidence for this relationship using newer longer records (Huang et al., 2012; Hu et al., 2012; García-Franco et al., 2022). Literature on observed QBO relationships with the tropical atmosphere (Lee et al., 2019), as well as literature on potential mechanisms, including a QBO-ENSO relationship, is also not cited (Nie and Sobel, 2015; García-Franco et al., 2023; Rodrigo et al., 2025).

2. **Mechanistic explanation not substantiated**

- The authors propose two routes of QBO influence: a Holton–Tan stratospheric pathway in DJF and a tropical UTLS–convection pathway in JJA. However, no direct diagnostics (e.g., tropopause temperature, OLR, wave activity fluxes) are provided to substantiate these claims. What is the main mechanism that relates the QBO to tropical convection? Stability, or shear, or both?

- The claim of "linear progression" of QBO impacts across phases is descriptive only. Without quantitative regression or symmetry diagnostics, this interpretation remains speculative.

3. **Treatment of ENSO is insufficient**

- The exclusion of "extreme" ENSO months using Niño-3.4 thresholds ($\pm 1$ K) is not, in my opinion, an adequate control. Previous studies have shown important aliasing between ENSO and the QBO that is not easily dealt with (Domeisen et al., 2019). Authors should at least show that in their remaining data, the sampling between ENSO and QBO phases is symmetrically distributed, so that readers know that the results are not simply due to having more El Niño's during a certain QBO phase.

- The manuscript mentions that similar results were obtained with a $\pm 0.5$ K cutoff, but these results are not shown. Authors should provide several figures to substantiate their claims.

- A more robust approach would be to regress out ENSO (and ideally other low-frequency modes such as IOD/PDO) and test the sensitivity of the results. Authors may follow Gray et al. (2018) for this approach. Without such checks, the attribution to QBO remains uncertain.

**Specific Comments**

1. **Role of the monsoon climatology section** The lengthy repetition of global monsoon climatology (largely reproducing Yoden et al., with only two additional years) adds little scientific value. If needed as a baseline, it should be explicitly framed as such and substantially condensed, or moved to Supplementary Information. References to review papers (Geen; Bordoni; Biasutti; Wang) would be more appropriate than re-presenting standard figures.

2. **Figures and color scales** The chosen precipitation colorbar is counterintuitive: green corresponds to negative anomalies and warm colors to positive anomalies. This convention clashes with intuitive and widely used palettes. A more standard sequential blue scale would aid interpretation.

3. **Terminology and wording** There are several odd or overstated phrases, such as "chronic convection" or L122-"definitive description of global monsoon rainfall patterns." I do not agree with the fact that GPCP is a definitive description of precipitation. It is one of many available products, all

of which have advantages and disadvantages. The overall wording should be revised to standard scientific expressions or rewritten to simpler expressions.

4. **Clarity of writing** Some sentences are difficult to understand (e.g., line 458: "But, again refining that the influence of the QBO is manifested at a regional scale where localized atmospheric circulations exist"). The intended meaning should be clarified and redundant phrasing removed. I advise revising the manuscript with this in mind.

5. **Hadley cells** The manuscript frequently suggests a relationship between their results and potential modulation through 'Hadley cells' or local meridional circulations. Perhaps the authors could provide more evidence of this by calculating more diagnostics about the Hadley cell, as in (Schwendike et al., 2014).

6. **Language consistency** The English throughout is uneven. For instance, "the deep convection" and "deep convection" are used interchangeably, and "QBO modulation" is sometimes written without the article. A careful language edit would improve readability, but comments here are made in a constructive spirit, recognizing that the authors may not be native speakers so this is merely a suggestion.

**References**

Biasutti, M., Voigt, A., Boos, W. R., Braconnot, P., Hargreaves, J. C., Harrison, S. P., Kang, S. M., Mapes, B. E., Scheff, J., Schumacher, C., et al. (2018). Global energetics and local physics as drivers of past, present and future monsoons. *Nature Geoscience*, 11(6):392–400.

Bordoni, S. and Schneider, T. (2008). Monsoons as eddy-mediated regime transitions of the tropical overturning circulation. *Nature Geoscience*, 1(8):515–519.

Domeisen, D. I., Garfinkel, C. I., and Butler, A. H. (2019). The teleconnection of el niño southern oscillation to the stratosphere. *Reviews of Geophysics*, 57(1):5–47.

García-Franco, J. L., Gray, L. J., Osprey, S., Chadwick, R., and Martin, Z. (2022). The tropical route of quasi-biennial oscillation (qbo) teleconnections in a climate model. *Weather and Climate Dynamics*, 3(3):825–844.

García-Franco, J. L., Gray, L. J., Osprey, S., Jaison, A. M., Chadwick, R., and Lin, J. (2023). Understanding the mechanisms for tropical surface impacts of the quasi-biennial oscillation (qbo). *Journal of Geophysical Research: Atmospheres*, 128(15):e2023JD038474.

Geen, R., Bordoni, S., Battisti, D. S., and Hui, K. (2020). Monsoons, itczs, and the concept of the global monsoon. *Reviews of Geophysics*, 58(4):e2020RG000700.

Gray, L. J., Anstey, J. A., Kawatani, Y., Lu, H., Osprey, S., and Schenzinger, V. (2018). Surface impacts of the Quasi Biennial Oscillation. *Atmospheric Chemistry and Physics*, 18(11):8227–8247.

Hu, Z.-Z., Huang, B., Kinter III, J. L., Wu, Z., and Kumar, A. (2012). Connection of the stratospheric qbo with global atmospheric general circulation and tropical sst. part ii: Interdecadal variations. *Climate dynamics*, 38(1):25–43.

Huang, B., Hu, Z.-Z., Kinter III, J. L., Wu, Z., and Kumar, A. (2012). Connection of stratospheric qbo with global atmospheric general circulation and tropical sst. part i: Methodology and composite life cycle. *Climate dynamics*, 38(1):1–23.

Lee, J.-H., Kang, M.-J., and Chun, H.-Y. (2019). Differences in the tropical convective activities at the opposite phases of the quasi-biennial oscillation. *Asia-Pacific Journal of Atmospheric Sciences*, 55(3):317–336.

Nie, J. and Sobel, A. H. (2015). Responses of tropical deep convection to the qbo: Cloud-resolving simulations. *Journal of the Atmospheric Sciences*, 72(9):3625–3638.

Rodrigo, M., García-Serrano, J., and Bladé, I. (2025). Quasi-biennial oscillation influence on tropical convection and el niño variability. *Geophysical Research Letters*, 52(10):e2024GL112854.

Schwendike, J., Govekar, P., Reeder, M. J., Wardle, R., Berry, G. J., and Jakob, C. (2014). Local partitioning of the overturning circulation in the tropics and the connection to the hadley and walker circulations. *Journal of Geophysical Research: Atmospheres*, 119(3):1322–1339.

Wang, B. and Ding, Q. (2008). Global monsoon: Dominant mode of annual variation in the tropics. *Dynamics of Atmospheres and Oceans*, 44(3-4):165–183.

---

## Referee Comment (RC2)

**General comments**

This study investigates stratospheric QBO teleconnections with global monsoon systems based on 42 years (1979–2020) of reanalysis data. By focusing on boreal summer (JJA) and austral summer (DJF), and by excluding extreme ENSO events, the authors examine how QBO anomalies in the lower stratosphere modulate global and regional atmospheric circulations and precipitation. The results highlight significant impacts over the Northwest Pacific, North Atlantic, and Northeast Pacific, where the QBO appears to influence circulation systems such as the Walker Circulation, the Azores High, and the PNA pattern, leading to distinct precipitation anomalies.

The topic is timely and of interest, and the scientific questions are well-posed. However, in its current form the manuscript does not yet meet the standards of completeness, clarity, and rigor required for publication. In particular, the underlying mechanisms of the proposed associations are not sufficiently demonstrated, and the manuscript requires substantial technical revisions. I therefore recommend major revisions before the manuscript can be considered for publication.

**Specific comments**

**Point1: Novelty of this study**

It remains unclear how this study fundamentally differs from Yoden et al. (2023), and what new findings are provided. Figures 1–9 appear very similar to those in Yoden et al. (2023), with the only difference being the inclusion of an additional two years in the analysis period. Specifically, Figs. 1–6 show the vertical profile of the QBO, followed by the climatological mean precipitation pattern and the impacts of phases 4–8 over the Northwest Pacific during JJA (Fig. 7), the impacts of phases 4–8 over the North Atlantic during DJF (Fig. 8), and the impacts of phases 5–1 over the Northeast Pacific during DJF (Fig. 9). These figures are nearly identical to those in Yoden et al. (2023). Figs. 10–11 show additional results, but these appear insufficient to elucidate the underlying mechanisms.

Thus, the novelty of this work requires clarification. The authors should explicitly specify 'what is new or improved.' For instance, this could involve identifying the impacts of the QBO in regions not considered by Yoden et al. (2023), or providing robust evidence of mechanisms through new model experiments or analyses not previously demonstrated, as you mentioned in lines 90–93.

*L90-93: "Recently Yoden et al. (2023) reported on new observational aspects of QBO modulation of the GM system, highlighting modulation of low-pressure cyclonic perturbations over the NH western Pacific during JJA and eastern Pacific during DJF. However, this study does not provide an in-depth view of the QBO association with the GM system from both a phenomenological and mechanical perspective."*

**Point 2: Causality and mechanisms**

The way in which zonal-mean QBO anomalies influence the regional GM system remains unclear. Why do the strongest impacts occur specifically over the Northwest Pacific, North Atlantic, and Northeast Pacific? Which atmospheric processes realize these linkages? A more concrete depiction and explanation would strengthen the manuscript. For example, time-lagged composites could demonstrate the temporal evolution of QBO-related signals and the development of circulation anomalies, or model experiments could help identify the key mechanisms in each region, even though establishing the exact causality of wave–mean flow interactions is inherently challenging.

Furthermore, the factors causing zonal-mean stratospheric signals to become zonally asymmetric in the troposphere, as well as their sensitivity to the QBO phase (50, 70 hPa), should be clarified. If these issues have already been addressed in previous studies, they should be discussed in greater detail in the Introduction. Currently, the Introduction provides only a broad overview of the tropical, subtropical, and stratospheric QBO teleconnection routes (L51–98). As in the ENSO literature, the documented regional impacts of the QBO, particularly in the Northwest Pacific, North Atlantic, and Northeast Pacific, and the limitations of current understanding should be presented more explicitly to the reader.

**Point3: ENSO definition and its impact**

A more detailed description of ENSO's impacts on the three target regions (Northwest Pacific, North Atlantic, Northeast Pacific) should be provided in the Introduction (L44–50). Does ENSO affect these regions independently of the QBO, or do they exert joint impacts? The rationale for excluding ENSO from the analysis should be explained more clearly to the reader.

*L44-50: 'On interannual time scales, the GM system is also influenced by the El Niño Southern Oscillation (ENSO), a major driver of global teleconnections (Mooley and Parthasarathy, 1984; Shen and Lau, 1995; Krishnamurthy and Goswamy, 2000; Yu et al., 2021). The impact of ENSO can be confined to a specific monsoon system and may vary with the time scale. Recently, Yu et al. (2021) demonstrated that, at interannual time scales, the Indian summer monsoon exhibits a stronger relationship with ENSO, whereas on the decadal time scale, this relationship is weaker. Despite extensive studies on the teleconnection between GM dynamics and surface atmospheric circulations, interannual variability of GM is still unclear, and further research is needed to improve understanding.'*

In this study, only ENSO events with amplitudes greater than |1.0 K| were excluded, which implies that moderate and weak ENSO events were included. The authors state that lowering the threshold to |0.5 K| yields the same results (L471–472). However, this raises the question of why all figures were not produced using the NOAA CPC threshold of |0.5 K|. If the intention was to re-examine the new teleconnection pathway reported by Kumar et al. (2024), the ENSO-neutral threshold of 0.4 K employed in that study would seem more appropriate. Instead, the threshold of 1.0 K used in Yoden et al. (2023) was adopted. This inconsistency is likely to raise questions among readers. A sufficient explanation of the rationale behind the chosen threshold should therefore be included.

*L471-472: 'Note that our analysis includes only the neutral ENSO phase, and obtained the same patterns even with a low threshold of the ENSO index (±0.5 K).'*

Additionally, the number of El Niño cases differs from Yoden et al. (2023). Although the analysis period was extended from 1979–2018 to 1979–2020 (two additional years), the number of El Niño months decreased from 59 to 57, while neutral and La Niña months increased. One would expect the El Niño count to remain stable or increase, so this discrepancy requires clarification. (374 neutral, 59 El Niño, and 47 La Niña → 395 neutral, 57 El Niño, and 52 La Niña).

**Point4: Completeness of the manuscript**

**4-1) Structure**

The structure of the main text should be revised. For example, Fig. 3 currently presents JJA U Difference (Fig. 3c) → JJA and DJF U (Figs. 3a, b, d, e; note there may also be a typo in the manuscript) → JJA and DJF T (Figs. 3g, h, j, k) for the thermal wind relationship → DJF U and T (Figs. 3l, f). What about Fig. 3i? Since the description mainly focuses on the differences (right panels), presenting only those panels might be enough.

In addition, the first two paragraphs of the Introduction could be merged into a single paragraph for better flow.

**4-2) References**

The appropriateness of the references should be carefully checked. For example, the manuscript states that increased midlatitude surface cold air outbreaks during QBO-E are reported by Kumar et al. (2022) (L61–64). However, Kumar et al. (2022) does not demonstrate surface cold air outbreaks. A more suitable reference should therefore be cited throughout the manuscript.

*L61-64: 'This pathway is known as "Holton-Tan effect", or H-T effect (Holton and Tan 1980; 1982), which operates during boreal winter when westerlies exist in the polar stratosphere, QBO E favors a disrupted polar vortex, and therefore a weak Northern Annular Mode (NAM) and increased midlatitude surface cold air outbreaks (e.g., Kumar et al., 2022).'*

**Minor comments**

1) It seems that P5–P1 shows a peak at 20 hPa rather than at 70 hPa (Figs. 1a,e). Could the authors clarify this?

*L165-167: 'this study will focus on the phases when QBO anomalies arrive in the UTLS regions, separately for JJA and DJF, with results presented for the composite difference P4 – P8 (QBO W – QBO E 50 hPa), as well as for P5 – P1 (QBO W – QBO E at 70 hPa)'.*

2) The STJs are not shown in the figures. Adding the climatological mean U (STJs) as contours would make the meridional shift more apparent. A similar modification is recommended for Fig. 4.

*L230-231: 'Along the UTLS, this dipole favors an equatorward shift of the zonal mean STJs during QBO W at 50 hPa'.*

3) Would 'reverse' be the correct term?

*L464: 'The reverse scenario during can be seen during the QBO E phase.'*

4) Are the years used to define QBO 50 hPa and QBO 70 hPa the same?

*L20: 'As QBO phase progress,'*

**Technical corrections**

There are numerous typographical errors, ranging from minor issues to those that may lead to misinterpretation of the results. These reduce the readability of the manuscript. Below I list only a subset. The authors are strongly advised to thoroughly proofread the entire manuscript to correct these errors and enhance clarity.

L585: The abstract and main text state that the analysis period is from 1979 to 2020. However, the

conclusions section refers to the period from 1979 to 2022.

L148: *profile of P1-P5 is different above 0.3 hPa* → 0.3?

L465: *Consistent with previous results (section 5.2 & 5.4)* → this sentence is written in section 5.4.

L489: *190°W and 260°W'* → 190-260E.

L14: *coincide with* → coincides with

L15: *Northen Hemisphere* → Northern Hemisphere

L15: *for same QBO W* → for the same QBO W

L20: *QBO phase progress* → QBO phase progresses

L67: *synoptic and 'and' planetary-scale* → remove 'and'

L101: *neither El Niño or La Niña* → neither El Niño nor La Niña

L102: *ESNO* → ENSO

L233: *Fig.s* → Figs.

L239: *(Figs., 3 f)*

L265: *P5 (hereinafter QBO W at 70h Pa) and P1 (hereinafter QBO W at 70 hPa)*

L497: *Brunt–väisälä frequency* → Brunt–Väisälä frequency

L505: *Zhou at el* → et al.

L581: *annualar mode* → annular mode

L613-614: *As a result, strengthens the anticyclonic circulations* → incomplete sentence..

---

## Author Comment (AC1)

The authors are thankful to the reviewer for providing valuable comments and suggestions to improve the quality of this paper. We have incorporated all comments to the maximum extent in the revised manuscript. A point by point reply is given below:

**Major Comments**

**1. Incomplete and selective literature review**

- The Introduction omits several key works on monsoons as well as QBO impacts on convection and precipitation.

  The authors frequently argue that the global monsoon is a land-sea contrast phenomena, overlooking recent literature on the monsoons as a results of energetics and solar forcing (Bordoni and Schneider, 2008; Biasutti et al., 2018; Geen et al., 2020), which include papers they cite on their manuscript (Wang and Ding, 2008). If the authors choose to delineate the global monsoon as the ocean-land thermal contrast only, they should at least mention other existing views and explain why they choose this approach. I realize this is only tangentially related to the whole manuscript but it's an important distinction.

  Then, the authors provide only a handful of references and insight into the tropical connection between the QBO and the surface. For example, they cite Yasunari correctly as a paper that relates the QBO to the Walker circulation strength but they fail to also mention more recent work that has found similar evidence for this relationship using newer longer records (Huang et al., 2012; Hu et al., 2012; Garc´ıa-Franco et al., 2022). Literature on observed QBO relationships with the tropical atmosphere (Lee et al., 2019), as well as literature on potential mechanisms, including a QBO-ENSO relationship, is also not cited (Nie and Sobel, 2015; Garc´ıa-Franco et al., 2023; Rodrigo et al., 2025).

  **Reply:** The introduction has been revised after incorporating all the above suggestions and citations. We believe that the revised introduction is complete and addresses the gap left by the omission of the literature review.

**Mechanistic explanation not substantiated**

- The authors propose two routes of QBO influence: a Holton–Tan stratospheric pathway in DJF and a tropical UTLS–convection pathway in JJA. However, no direct diagnostics (e.g., tropopause temperature, OLR, wave activity fluxes) are provided to substantiate these claims. What is the main mechanism that relates the QBO to tropical convection? Stability, or shear, or both?

  **Reply:** We propose that the UTLS pathway operates during all seasons and that the H-T mechanism does not operate in summer. For the UTLS pathway we argue that QBO MMC temperature anomalies influence convective regions, which then affect local precipitation and radiation of planetary wave trains along a great circle route. We show effects on temperature, geopotential height, winds, and precipitation which are consistent with QBO modulation of the PNA and NAO patterns via this mechanism. We do not show wave activity flux. However, the teleconnection argument makes use of known mechanisms: modulation of tropical convective centers, modulation of Rossby wave source, modulation of Rossby wave train, modulation of extratropical centers of action. We are discussing statistically significant examples of this process. We show that the summer QBO circulation anomaly is likely related to the MMC temperature cold anomaly in the subtropics, which is the UTLS pathway, and that the H-T pathway does not exist in the summer. We cannot determine during boreal winter what the relative contributions of the UTLS pathway and the H-T / annular mode mechanism. The main mechanism that is emerging is that UTLS temperature anomalies are important. In the tropics there is a coincidence of reduced shear and cold anomaly geographically during QBO E (Hitchman et al. 2021), so they tend to co-vary.

  As part of our continuing research, we are conducting three separate detailed follow-up studies focusing on individual regions to explore the underlying mechanisms in greater depth.

- The claim of "linear progression" of QBO impacts across phases is descriptive only. Without quantitative regression or symmetry diagnostics, this interpretation remains speculative.

  **Reply:** We find that the anomaly patterns are similar but of the opposite sign for QBO E and QBO W for the statistically significant circulation anomalies that we are focusing on in this study. This is consistent with a linear relationship between QBO phase and phase of response in the teleconnection. However, you are right that we did not demonstrate any sort of "progression", rather that the two end points are consistent with linearity, so we removed "progression".

**3. Treatment of ENSO is insufficient**

- The exclusion of "extreme" ENSO months using Niño-3.4 thresholds (±1 K) is not, in my opinion, an adequate control. Previous studies have shown important aliasing between ENSO and the QBO that is not easily dealt with (Domeisen et al., 2019). Authors should at least show that in their remaining data, the sampling between ENSO and QBO phases is symmetrically distributed, so that readers know that the results are not simply due to having more El Niño's during a certain QBO phase.
  **Reply:** We appreciate your concern regarding the possible influence of ENSO bias. In consistency with our previous studies (Hitchman et al., 2020; Kumar et al., 2022, 2024; Yoden et al., 2023), we adopted a simple approach of calculating direct composite differences between the QBO opposite phases. Any month has been excluded from QBO W and E composites for Niño 3.4 index thresholds (±1K). Using the same threshold value in each composite for evaluating ENSO neutral conditions helps to avoid El Niño related biases during a certain QBO phase.

- The manuscript mentions that similar results were obtained with a ±0.5 K cutoff, but these results are not shown. Authors should provide several figures to substantiate their claims.
  **Reply:** In response to the concern raised by the other reviewer and to maintain consistency with our previous study (Kumar et al., 2024), we have adopted a cutoff value of ±0.4 K. The corresponding figures are provided as supplementary information wherever necessary. For instance, when discussing the possible mechanisms for QBO – PNA teleconnection, Fig. S3 has been included for the ±0.4 K case corresponding to Fig. 9. We find that the patterns which we discuss are very similar using the lower threshold.

- A more robust approach would be to regress out ENSO (and ideally other low-frequency modes such as IOD/PDO) and test the sensitivity of the results. Authors may follow Gray et al. (2018) for this approach. Without such checks, the attribution to QBO remains uncertain.
  **Reply:** The multivariate regression of QBO W–E differences was the main technique employed by Gray et al. (2018). Their sensitivity tests showed that the QBO regression coefficients were unaffected by the inclusion or exclusion of solar, volcanic, ENSO, and trend terms. Their tests also confirmed that the results were essentially the same for both the ERA-Interim period since 1979 and the entire period from 1958 to 2016. We have included this discussion in the revised introduction to address the possible ENSO bias in the QBO influence. In this study, we have directly excluded ENSO events from each QBO composite phase using a threshold value to avoid the ENSO impact on the QBO influence.

**Specific Comments:**

1. **Role of the monsoon climatology section** The lengthy repetition of global monsoon climatology (largely reproducing Yoden et al., with only two additional years) adds little scientific value. If needed as a baseline, it should be explicitly framed as such and substantially condensed, or moved to Supplementary Information. References to review papers (Geen; Bordoni; Biasutti; Wang) would be more appropriate than re-presenting standard figures.
   **Reply:** To facilitate easy understanding of the geographical distribution of the seasonal mean basic state quantities, we use this figure as a reference when discussing the longitude–latitude sections of QBO composite differences in precipitation and horizontal winds (Fig. 5). Therefore, it is more relevant to keep this climatology section in the main text. However, as per your suggestion, we have shortened the discussion on global monsoon climatology, retaining only the essential content. Also, we have incorporated the suggested references.

2. **Figures and color scales** The chosen precipitation colorbar is counterintuitive: green corresponds to negative anomalies and warm colors to positive anomalies. This convention clashes with intuitive and widely used palettes. A more standard sequential blue scale would aid interpretation.
   **Reply:** We completely value your viewpoint. However, for the sake of consistency with our previous publications (Kumar et al., 2022; 2024; Yoden et al., 2023), we would like to retain the current color bar. This color bar was originally chosen to maintain consistency with certain standard references, including the Global Monsoon System series edited by C.-P. Chang. Please allow us to use the current color bar.

3. **Terminology and wording** There are several odd or overstated phrases, such as "chronic convection" or L122-"definitive description of global monsoon rainfall patterns." I do not agree with the fact that GPCP is a definitive description of precipitation. It is one of many available products, all of which have advantages and disadvantages. The overall wording should be revised to standard scientific expressions or rewritten to simpler expressions.

**Reply:** Ok. We have revised the sentence as "This dataset provides an overview of global monsoon rainfall patterns over both land and ocean". We have also refined the text where necessary to ensure smooth and clear scientific expression. In the case of "chronic convecction", the Webster-Miriam dictionary defines chronic as continuing or occurring again and again for a long time. This is what we mean by a region where there is usually deep convection during a given season. But perhaps there is some issue with the fact that people sometimes use chronic to refer to an illness. No, we are not trying to say that the atmosphere is ill, of course! But to allay such a concern we will refer to climatological areas of deep convection, as in "These warm anomalies coincide with the climatological locations of deep convection, characterized by cloud top temperatures less than 192 K and low OLR emission (Collimore et al., 1998)" on new lines 380-381.

**Clarity of writing** Some sentences are difficult to understand (e.g., line 458: "But, again refining that the influence of the QBO is manifested at a regional scale where localized atmospheric circulations exist"). The intended meaning should be clarified and redundant phrasing removed. I advise revising the manuscript with this in mind.

**Reply:** The language of the manuscript has been carefully revised to enhance its clarity and readability.

4. **Hadley cells** The manuscript frequently suggests a relationship between their results and potential modulation through 'Hadley cells' or local meridional circulations. Perhaps the authors could provide more evidence of this by calculating more diagnostics about the Hadley cell, as in (Schwendike et al., 2014).

**Reply:** Yes, we agree that referring to the modulation of the Hadley cell is not appropriate based on the current analysis. We have removed reference to local Hadley cells. Instead, we aim to highlight the changes in upward motion and their association with the strength of the Bonin High. In the revised manuscript, we have updated the text to read as follows:

L549- L555 "Another aspect of this system which may be important in this context is that deep tropical convection is suppressed over Indonesia during QBO W, which implies reduced subtropical downwelling. The lower tropospheric anticyclone which dominates the circulation east of Japan during summer (the Bonin High, Enomoto et al., 2003) is one region of subsidence. With reduced convection in the adjacent tropics, one might expect a reduction in subsidence and strength of the Bonin High. In our analysis, the reduced subtropical downwelling is evident as a significant upward W-wind anomaly accompanied by notable meridional wind contours, suggesting a weakening of the region of subsidence over east of Japan (Fig. 10c, d)."

Thanks for the suggesting the nice reference (Schwendike et al., 2014), which has been incorporated into the manuscript to describe the climatological vertical motion field during DJF when discussing the mechanism for QBO modulation in of the North Atlantic Ocean (L567-570).

5. **Language consistency** The English throughout is uneven. For instance, "the deep convection" and "deep convection" are used interchangeably, and "QBO modulation" is sometimes written without the article. A careful language edit would improve readability, but comments here are made in a constructive spirit, recognizing that the authors may not be native speakers so this is merely a suggestion.

**Reply:** As replied in the above comment, the language of the manuscript has been carefully revised to enhance its clarity and readability.

**References**

Biasutti, M., Voigt, A., Boos, W. R., Braconnot, P., Hargreaves, J. C., Harrison, S. P., Kang, S. M., Mapes, B. E., Scheff, J., Schumacher, C., et al. (2018). Global energetics and local physics as drivers of past, present and future monsoons. *Nature Geoscience*, 11(6):392–400.

Bordoni, S. and Schneider, T. (2008). Monsoons as eddy-mediated regime transitions of the tropical overturning circulation. *Nature Geoscience*, 1(8):515–519.

Domeisen, D. I., Garfinkel, C. I., and Butler, A. H. (2019). The teleconnection of el niño southern oscillation to the stratosphere. *Reviews of Geophysics*, 57(1):5–47.

Garc´ıa-Franco, J. L., Gray, L. J., Osprey, S., Chadwick, R., and Martin, Z. (2022). The tropical route of quasi-biennial oscillation (qbo) teleconnections in a climate model. *Weather and Climate Dynamics*, 3(3):825–844.

Garc´ıa-Franco, J. L., Gray, L. J., Osprey, S., Jaison, A. M., Chadwick, R., and Lin, J. (2023). Understanding the mechanisms for tropical surface impacts of the quasi-biennial oscillation (qbo). *Journal of Geophysical Research: Atmospheres*, 128(15):e2023JD038474.

Geen, R., Bordoni, S., Battisti, D. S., and Hui, K. (2020). Monsoons, itczs, and the concept of the global monsoon. *Reviews of Geophysics*, 58(4):e2020RG000700.

Gray, L. J., Anstey, J. A., Kawatani, Y., Lu, H., Osprey, S., and Schenzinger, V. (2018). Surface impacts of the Quasi Biennial Oscillation. *Atmospheric Chemistry and Physics*, 18(11):8227–8247.

Hu, Z.-Z., Huang, B., Kinter III, J. L., Wu, Z., and Kumar, A. (2012). Connection of the stratospheric qbo with global atmospheric general circulation and tropical sst. part ii: Interdecadal variations. *Climate dynamics*, 38(1):25–43.

Huang, B., Hu, Z.-Z., Kinter III, J. L., Wu, Z., and Kumar, A. (2012). Connection of stratospheric qbo with global atmospheric general circulation and tropical sst. part i: Methodology and composite life cycle. *Climate dynamics*, 38(1):1–23.

Lee, J.-H., Kang, M.-J., and Chun, H.-Y. (2019). Differences in the tropical convective activities at the opposite phases of the quasi-biennial oscillation. *Asia-Pacific Journal of Atmospheric Sciences*, 55(3):317–336.

Nie, J. and Sobel, A. H. (2015). Responses of tropical deep convection to the qbo: Cloud-resolving simulations. *Journal of the Atmospheric Sciences*, 72(9):3625–3638.

Rodrigo, M., Garc´ıa-Serrano, J., and Bladé, I. (2025). Quasi-biennial oscillation influence on tropical convection and el niño variability. *Geophysical Research Letters*, 52(10):e2024GL112854.

Schwendike, J., Govekar, P., Reeder, M. J., Wardle, R., Berry, G. J., and Jakob, C. (2014). Local partitioning of the overturning circulation in the tropics and the connection to the hadley and walker circulations. *Journal of Geophysical Research: Atmospheres*, 119(3):1322–1339.

Wang, B. and Ding, Q. (2008). Global monsoon: Dominant mode of annual variation in the tropics. *Dynamics of Atmospheres and Oceans*, 44(3-4):165–183

**Reply:** We have cited all above suggested references at appropriate place in the revised manuscript.

---

## Author Comment (AC2)

The authors are thankful to the reviewer for providing valuable comments and suggestions to improve the quality of this paper. We have incorporated all comments to the maximum extent in the revised manuscript. A point by point reply is given below:

**General comments**

This study investigates stratospheric QBO teleconnections with global monsoon systems based on 42 years (1979–2020) of reanalysis data. By focusing on boreal summer (JJA) and austral summer (DJF), and by excluding extreme ENSO events, the authors examine how QBO anomalies in the lower stratosphere modulate global and regional atmospheric circulations and precipitation. The results highlight significant impacts over the Northwest Pacific, North Atlantic, and Northeast Pacific, where the QBO appears to influence circulation systems such as the Walker Circulation, the Azores High, and the PNA pattern, leading to distinct precipitation anomalies.

The topic is timely and of interest, and the scientific questions are well-posed. However, in its current form the manuscript does not yet meet the standards of completeness, clarity, and rigor required for publication. In particular, the underlying mechanisms of the proposed associations are not sufficiently demonstrated, and the manuscript requires substantial technical revisions. I therefore recommend major revisions before the manuscript can be considered for publication.

**Reply:** Thank you for evaluating the manuscript and for your constructive feedback on the overall quality improvement of the paper. We have revised the manuscript to meet the standards of completeness, clarity, and rigor required for publication. We hope that the revised version provides a more comprehensive and coherent discussion on the underlying mechanisms. All technical errors have been corrected as per your suggestions, including additional corrections made in other relevant places.

**Specific comments**

**Point1: Novelty of this study**

It remains unclear how this study fundamentally differs from Yoden et al. (2023), and what new findings are provided. Figures 1–9 appear very similar to those in Yoden et al. (2023), with the only difference being the inclusion of an additional two years in the analysis period. Specifically, Figs. 1– 6 show the vertical profile of the QBO, followed by the climatological mean precipitation pattern and the impacts of phases 4–8 over the Northwest Pacific during JJA (Fig. 7), the impacts of phases 4–8 over the North Atlantic during DJF (Fig. 8), and the impacts of phases 5–1 over the Northeast Pacific during DJF (Fig. 9). These figures are nearly identical to those in Yoden et al. (2023). Figs. 10– 11 show additional results, but these appear insufficient to elucidate the underlying mechanisms.

Thus, the novelty of this work requires clarification. The authors should explicitly specify 'what is new or improved.' For instance, this could involve identifying the impacts of the QBO in regions not considered by Yoden et al. (2023), or providing robust evidence of mechanisms through new model experiments or analyses not previously demonstrated, as you mentioned in lines 90–93.

L90-93: "Recently Yoden et al. (2023) reported on new observational aspects of QBO modulation of the GM system, highlighting modulation of low-pressure cyclonic perturbations over the NH western Pacific during JJA and eastern Pacific during DJF. However, this study does not provide an in-depth view of the QBO association with the GM system from both a phenomenological and mechanical perspective."

**Reply:** Thank you for your concern. It is indeed essential to highlight the novelty of any work in comparison with the existing literature. The introduction has been revised after including the more citations. We believed that revised introduction smoothly expresses the novel aspect of this paper.

First, Yoden et al. (2023) primarily focused on presenting the long-term climatology of global monsoon systems and associated circulations during the satellite era (1979 onward) but provides only a brief overview on the QBO modulations. The analysis in the present study is substantially different from Yoden et al. (2023).

Although Figure 1 appears similar to the corresponding figure in Yoden et al. (2023), it represents an updated version covering a wider latitudinal domain (60°S–60°N) and includes wind patterns at lower the boundary layer 950 hPa. It is not emphasized as part of new results, but is used to lay the geographical context for major features of the seasonal monsoon system. Figure 2 discusses the differences in the seasonal structure of the QBO at different levels, whereas the corresponding figure in Yoden et al. (2023) illustrates the common QBO structure based on year-round data. These figures form the basis of the present analysis and hence cannot be omitted. Figures 3 and 4 present entirely new results derived from analyses of zonal-mean quantities. Figure 5 provides a new and more compact summary of Figures 7–10 in Yoden et al. (2023) for extended broader domain. Figures 6–9 present new results that were neither discussed nor included in the previous study. They constitute the main findings of this work and provide detailed phenomenological description. The limitation of Yoden et al. (2023) served as a primary motivation for the present study.

**Point 2: Causality and mechanisms**

The way in which zonal-mean QBO anomalies influence the regional GM system remains unclear. Why do the strongest impacts occur specifically over the Northwest Pacific, North Atlantic, and Northeast Pacific? Which atmospheric processes realize these linkages? A more concrete depiction and explanation would strengthen the manuscript. For example, time-lagged composites could demonstrate the temporal evolution of QBO-related signals and the development of circulation anomalies, or model experiments could help identify the key

mechanisms in each region, even though establishing the exact causality of wave–mean flow interactions is inherently challenging.

**Reply:** We fully acknowledge your valuable suggestion that time-lagged composites could demonstrate the temporal evolution of QBO-related signals and the development of circulation anomalies. However, the present study was designed to provide a first-step global perspective to identify and highlight the key regions with possible mechanism where the QBO signal is most evident in the monsoon and circulation systems. The current paper is already substantial lengthy, and inclusion of further additional analysis would make the manuscript excessively long and may deviate from its primary objective of establishing the global context.

As part of our continuing research, we are conducting detailed follow-up studies focusing on individual regions to explore the underlying mechanisms in greater depth. We will test the time-lagged composites in these studies for temporal evolution of QBO-related signals.

Furthermore, the factors causing zonal-mean stratospheric signals to become zonally asymmetric in the troposphere, as well as their sensitivity to the QBO phase (50, 70 hPa), should be clarified. If these issues have already been addressed in previous studies, they should be discussed in greater detail in the Introduction.

Currently, the Introduction provides only a broad overview of the tropical, subtropical, and stratospheric QBO teleconnection routes (L51–98). As in the ENSO literature, the documented regional impacts of the QBO, particularly in the Northwest Pacific, North Atlantic, and Northeast Pacific, and the limitations of current understanding should be presented more explicitly to the reader.

**Reply:** The zonally symmetric QBO UTLS temperature anomalies can cause an asymmetric tropospheric response through their influence on deep convection in certain regions. The MMCs descend with time. Progressing from the 50 hPa to the 70 hPa index, the QBO thermal anomalies are lower in altitude within the UTLS and have a different effect. During JJA convection is centered over southeast Asia, off of the equator. It is not clear yet exactly why there is a big difference between using the 50 and 70 hPa indices, but the tropopause is much lower over the Maritime Continent during JJA compared to DJF, so it is more likely that the warm anomaly associated with the 70 hPa index will suppress convection over the equator. In the Northeast Pacific case both indices show reduced Bonin High and northeastward-extended South Asian High, compatible with a zonal mean cold anomaly in the subtropics, which locally enhances the spread of convection over the Northwest Pacific.

The introduction and discussion have been revised after considering the above suggestions. We believe that the revised version provides clear information on both the sensitivity to the QBO phase and the regional impacts of ENSO across the three targeted regions. Furthermore, the main text also discusses the possible factors responsible for the zonal-mean stratospheric signals becoming zonally asymmetric in the troposphere.

**Point3: ENSO definition and its impact**

A more detailed description of ENSO's impacts on the three target regions (Northwest Pacific, North Atlantic, Northeast Pacific) should be provided in the Introduction (L44–50). Does ENSO affect these regions independently of the QBO, or do they exert joint impacts? The rationale for excluding ENSO from the analysis should be explained more clearly to the reader.

L44-50: 'On interannual time scales, the GM system is also influenced by the El Niño Southern Oscillation (ENSO), a major driver of global teleconnections (Mooley and Parthasarathy, 1984; Shen and Lau, 1995; Krishnamurthy and Goswamy, 2000; Yu et al., 2021). The impact of ENSO can be confined to a specific monsoon system and may vary with the time scale. Recently, Yu et al. (2021) demonstrated that, at interannual time scales, the Indian summer monsoon exhibits a stronger relationship with ENSO, whereas on the decadal time scale, this relationship is weaker. Despite extensive studies on the teleconnection between GM dynamics and surface atmospheric circulations, interannual variability of GM is still unclear, and further research is needed to improve understanding.'

**Reply:** As mentioned in the above comment, the introduction section has been revised to include a more detailed description of ENSO's impacts on the three target regions. It also provides information on the rationale for excluding ENSO from the analysis.

In this study, only ENSO events with amplitudes greater than |1.0 K| were excluded, which implies that moderate and weak ENSO events were included. The authors state that lowering the threshold to |0.5 K| yields the same results (L471–472). However, this raises the question of why all figures were not produced using the NOAA CPC threshold of |0.5 K|. If the intention was to re-examine the new teleconnection pathway reported by Kumar et al. (2024), the ENSO-neutral threshold of 0.4 K employed in that study would seem more appropriate. Instead, the threshold of 1.0 K used in Yoden et al. (2023) was adopted. This inconsistency is likely to raise questions among readers. A sufficient explanation of the rationale behind the chosen threshold should therefore be included.

L471-472: 'Note that our analysis includes only the neutral ENSO phase, and obtained the same patterns even with a low threshold of the ENSO index (±0.5 K).'

**Reply:** In this study, the QBO is divided into eight 45° angular bin phases, and the sample size varies with phase and season. Some phases have a limited sample size, and lowering the threshold to 0.5K would further

reduce the sample size for certain phases; therefore, we have not shown the figures for this threshold in the main text. However, in view of your concern and to maintain consistency with our previous studies (Kumar et al., 2022; 2024), we have included Fig. S3 (in the supplementary information) for the ±0.4 K cutoff case corresponding to Fig. 9 when discussing mechanisms based on Kumar et al. (2024).

Additionally, the number of El Niño cases differs from Yoden et al. (2023). Although the analysis period was extended from 1979–2018 to 1979–2020 (two additional years), the number of El Niño months decreased from 59 to 57, while neutral and La Niña months increased. One would expect the El Niño count to remain stable or increase, so this discrepancy requires clarification. (374 neutral, 59 El Niño, and 47 La Niña ⬜ 395 neutral, 57 El Niño, and 52 La Niña).

**Reply:** Thank you very much for the in-depth study of the paper. Since ENSO events are also classified according to the QBO phases. Therefore, the phase angle plays a crucial role in counting El Niño, La Niña, and Neutral events. Additionally, a QBO state is defined using de-seasonalized zonal-mean zonal wind variations. The minor change in the climatology cycle value due to the inclusion of two additional years and the upgradation of data version (from ERA-Interim to ERA-5) induced small perturbations in the variance of the leading EOF1 and EOF2. For instance, the variances of EOF1 and EOF2 change from 59.7% and 34.9% (total 94.6%) in Yoden et al. (2023 to 58.1% and 36.7% (total 94.8%) in the current study, respectively. Consequently, there are minor changes in the QBO phase angle.

**Point4: Completeness of the manuscript**

**4-1) Structure:**

The structure of the main text should be revised. For example, Fig. 3 currently presents JJA U Difference (Fig. 3c) → JJA and DJF U (Figs. 3a, b, d, e; note there may also be a typo in the manuscript) → JJA and DJF T (Figs. 3g, h, j, k) for the thermal wind relationship → DJF U and T (Figs. 3l, f). What about Fig. 3i? Since the description mainly focuses on the differences (right panels), presenting only those panels might be enough.

**Reply:** We have revised both Figs. 3 and 4, per your suggestion.

In addition, the first two paragraphs of the Introduction could be merged into a single paragraph for better flow.

**Reply:** Done.

**4-2) References**

The appropriateness of the references should be carefully checked. For example, the manuscript states that increased midlatitude surface cold air outbreaks during QBO-E are reported by Kumar et al. (2022) (L61–64). However, Kumar et al. (2022) does not demonstrate surface cold air outbreaks. A more suitable reference should therefore be cited throughout the manuscript.

L61-64: 'This pathway is known as "Holton-Tan effect", or H-T effect (Holton and Tan 1980; 1982), which operates during boreal winter when westerlies exist in the polar stratosphere, QBO E favors a disrupted polar vortex, and therefore a weak Northern Annular Mode (NAM) and increased midlatitude surface cold air outbreaks (e.g., Kumar et al., 2022).'

**Reply:** Thanks for your in-depth review. The error in the year was typographical and has been corrected to Kumar et al. (2024), and also adding more new appropriate references to support this statement.

**Minor comments**

**1)** It seems that P5–P1 shows a peak at 20 hPa rather than at 70 hPa (Figs. 1a, e). Could the authors clarify this?

L165-167: 'this study will focus on the phases when QBO anomalies arrive in the UTLS regions, separately for JJA and DJF, with results presented for the composite difference P4 – P8 (QBO W – QBO E 50 hPa), as well as for P5 – P1 (QBO W – QBO E at 70 hPa)'.

**Reply:** The P5–P1 corresponds to QBO W – QBO E at 50 hPa and simultaneously opposite phase at 20 hPa with the peak magnitude. This can be more clearly seen in the revised Fig. 4 a and b. We have clarified the above point in the revised manuscript.

The STJs are not shown in the figures. Adding the climatological mean U (STJs) as contours would make the meridional shift more apparent. A similar modification is recommended for Fig. 4.

L230-231: 'Along the UTLS, this dipole favors an equatorward shift of the zonal mean STJs during QBO W at 50 hPa.'

**Reply:** As per your suggestion, we have added the climatological mean zonal wind (U) as contours on both Figs. 3 and 4. Thanks for nice suggestion.

**2)** Would 'reverse' be the correct term?

L464: 'The reverse scenario during can be seen during the QBO E phase.'

**Reply:** We have revised the sentence as "The opposite effects can be seen during the QBO E phase."

**3)** Are the years used to define QBO 50 hPa and QBO 70 hPa the same?
L20: 'As QBO phase progress,'

**Reply:** The same range of years was used, but the particular months chosen for W or E binning depend on the level. But the word "progresses" was misleading. We have omitted the "phase progress" in the revised manuscript.

**Technical corrections**
There are numerous typographical errors, ranging from minor issues to those that may lead to misinterpretation of the results. These reduce the readability of the manuscript. Below I list only a subset. The authors are strongly advised to thoroughly proofread the entire manuscript to correct these errors and enhance clarity.

**Reply:** Thank you for the critical reading and valuable suggestions to improve the quality of the paper. We have carefully proofread the entire manuscript to remove numerous typographical and minor errors.

L585: The abstract and main text state that the analysis period is from 1979 to 2020. However, the conclusions section refers to the period from 1979 to 2022.

**Reply:** The analysis period, 1979–2020, has been corrected in relevant places.

L148: profile of P1-P5 is different above 0.3 hPa →0.3?
**Reply:** Typographical error, and corrected 3 hPa.

L465: Consistent with previous results (section 5.2 & 5.4) →  this sentence is written in section 5.4
**Reply:**  Again, the typographical error; the reference has been corrected to "sections 5.2 & 5.3".

5.4. L489:190°W and 260°W'→190-260E.
L14: coincide with → coincides with
L15: Northen Hemisphere → Northern Hemisphere
L15: for same QBO W →  for the same QBO W
L20: QBO phase progress →  QBO phase progresses
L67: synoptic and 'and' planetary-scale → remove 'and'
L101: neither El Niño or La Niña → neither El Niño nor La Niña
L102: ESNO → ENSO
L233: Fig.s → Figs. L239: (Figs., 3 f)
**Reply:**  Thank you. All the above noted errors have been corrected in the revised manuscript.

L265: P5 (hereinafter QBO W at 70h Pa) and P1 (hereinafter QBO W at 70 hPa)
**Reply:** The text inside the parentheses associated with P1 has been corrected as "(hereinafter QBO E at 70 hPa)".

L497: Brunt–väisälä frequency → Brunt–Väisälä frequency
L505: Zhou at el → et al.
L581: annualar mode → annular mode
**Reply:**  Thank you, Corrected

L613-614: As a result, strengthens the anticyclonic circulations → incomplete sentence.
**Reply:** We have revised the sentence as "QBO W at 70 hPa promotes a positive PNA phase, which intensifies the mid-latitude northeastward flow over the Pacific Ocean".